# A Framework for Diffusion-Based Maximum Entropy Reinforcement Learning Algorithms

## Abstract

Diffusion models have shown strong performance in sampling from complex, unnormalized distributions. Building on this, we reinterpret Maximum Entropy Reinforcement Learning (ME-RL) as a diffusion-based sampling problem. We minimize the reverse KL divergence between a diffusion policy and the optimal policy distribution via a tractable upper bound, and apply the policy gradient theorem to derive a modified surrogate objective that incorporates diffusion dynamics through an augmented MDP and a diffusion-aware Q-function. This yields diffusion-based variants of Proximal Policy Optimization (PPO), Wasserstein Policy Optimization (WPO), and Relative Entropy Pathwise Policy Optimization (REPPO), which we term DME-PPO, DME-WPO, and DME-REPPO. These methods require only minor implementation changes to their base algorithms and, crucially, avoid backpropagation through the full diffusion chain. Experiments on standard continuous-control benchmarks demonstrate strong performance and validate Diffusion-Based Maximum Entropy Reinforcement Learning (DME-RL) as a practical, effective framework.

## 1. Introduction

Diffusion models (Song et al., 2021a; Sohl-Dickstein et al., 2015; Ho et al., 2020) have rapidly emerged as one of the most powerful tools for generative modeling, achieving state-of-the-art performance across image, video, and trajectory synthesis (Rombach et al., 2022; Ho et al., 2022; Chi et al., 2025). Beyond generative AI, recent work has begun applying diffusion samplers to *sampling from unnormalized target distributions*.

[1]Anonymous Institution, Anonymous City, Anonymous Region, Anonymous Country. Correspondence to: Anonymous Author <anon.email@domain.com>.

Preliminary work. Under review by the International Conference on Machine Learning (ICML). Do not distribute.

In this setting, one aims to sample from a distribution in the form of $\pi(x) = \frac{\exp(-\alpha E(x))}{\mathcal{Z}}$ where $\mathcal{Z} = \int_{\mathbb{R}^N} \exp(-\alpha E(x))\, dx$, $x \in \mathbb{R}^N$, $E(x)$ is an energy function and $\alpha = \frac{1}{\mathcal{T}}$ where $\mathcal{T} \geq 0$ is the temperature. Only the unnormalized density $\tilde{\pi}(x) := \exp(-\alpha E(x))$ is evaluatable, while exact sampling and the computation of the normalizing constant (partition function) $\mathcal{Z}$ is intractable. Importantly, there is no available data from this distribution. Diffusion samplers provide an expressive solution to this problem by transporting noise samples toward high-density regions of $\pi$ via a learned reverse diffusion process. These approaches have been explored in both the continuous domain (Zhang & Chen, 2022; Berner et al., 2022; Vargas et al., 2023; 2024; Richter & Berner, 2024) and the discrete domain (Sanokowski et al., 2024; 2025a), demonstrating strong performance on challenging sampling problems.

In the context of reinforcement learning, diffusion models are particularly appealing because they can represent complex, multimodal action distributions, which are often encountered in decision-making problems. Even when the optimal action distribution is unimodal, diffusion models provide a flexible mechanism to capture non-Gaussian shapes, such as heavy-tailed or Laplace-like distributions, allowing for more expressive and accurate policy representations than standard Gaussian approximations. Thus, diffusion models might be a promising approach to improve exploration, robustness, and ultimately performance on challenging RL environments.

Reinforcement learning naturally fits the unnormalized-sampling paradigm. A long line of work has shown that RL can be formulated as probabilistic inference (Todorov, 2008; Ziebart, 2010; Kappen et al., 2012; Levine, 2018) and thus RL corresponds to sampling from an unnormalized distribution —precisely the regime where diffusion samplers excel.

**Our approach:** In this work, we extend the sampling perspective of Maximum Entropy Reinforcement Learning (ME-RL) to diffusion models (see Sec. 2 and Sec. 3). Specifically, we introduce three diffusion-augmented Maximum Entropy RL algorithms: **DME-PPO**, **DME-REPPO**, and **DME-WPO**. Each algorithm arises from a shared reverse-KL formulation and is obtained by augmenting an existing

RL algorithm with a diffusion policy that samples actions through a learned reverse diffusion process. Concretely, **DME-REPPO** is a diffusion extension of Relative Entropy Pathwise Policy Optimization (Voelcker et al., 2025), **DME-PPO** extends Proximal Policy Optimization (PPO) (Schulman et al., 2017), and **DME-WPO** is a diffusion-based variant of Maximum Entropy Wasserstein Policy Optimization (ME-WPO), which generalizes Wasserstein Policy Optimization (WPO) (Pfau et al., 2025) to ME-RL. Our proposed framework can also be viewed as a generalization of Sanokowski et al. (2025a), who train diffusion models using RL, whereas here RL itself is performed *through* diffusion.

Diffusion models have recently been applied to ME-RL, but existing methods face some limitations. Many rely on forward-KL objectives (Dong et al., 2025; Ma et al., 2025) combined with importance weighting, which introduces bias, high variance, and *mode-covering* (Minka et al., 2005) behavior misaligned with the Reinforcement Learning goal of achieving the best average return. Our DME-RL algorithms avoid possibly memory-intensive backpropagation through the whole diffusion chain, unlike DiME (Celik et al., 2025). Meanwhile, DME-PPO method generalizes DPPO (Ren et al., 2025) to arbitrary temperatures, reducing to DPPO exactly when $\mathcal{T} = 0$. For a detailed discussion, see App. A.2.

In summary, by formulating RL as sampling from a target distribution over a sequence of actions. By using the data processing inequality, we derive a tractable upper bound on the reverse KL divergence between a diffusion policy and the target distribution (see Sec. 3). This yields a clean theoretical connection between diffusion models and ME-RL, which we coin *Diffusion-based Maximum Entropy Reinforcement Learning* (DME-RL) and leads naturally to the DME-PPO, DME-REPPO, and DME-WPO algorithms with minimal implementation overhead by small modifications of the corresponding MDP, the surrogate objective, and the value function (see Sec. 3.1).

**Our main contributions are as follows:**

- We introduce a unified diffusion-based ME-RL framework derived from reverse KL minimization (Sec. 3), yielding a tractable surrogate objective and a diffusion-augmented MDP and Q-function.

- Building on this framework, we instantiate three practical algorithms: **DME-PPO**, **DME-REPPO**, and **DME-WPO**, which require only minor modifications to their base algorithm and avoid backpropagation through the full diffusion chain.

- To derive **DME-WPO**, we first introduce **ME-WPO**, a maximum entropy generalization of Wasserstein Policy Optimization (WPO) in Sec. 2.3 and show in Sec. 4 that ME-WPO outperforms WPO.

- Our experiments in Sec. 4 show that DME-RL-based algorithms achieve strong performance on popular continuous-control benchmarks.

- Beyond the core algorithms, we establish in Sec. 2.2 a new connection between the off-policy ME-RL surrogate loss (e.g., used in **SAC** (Haarnoja et al., 2018)) and the Log-Variance/Trajectory-Balance loss (Richter et al., 2020; Malkin et al., 2022a) commonly used in Diffusion Samplers or GFlow Networks (Bengio et al., 2021a).

## 2. Problem Description

Reinforcement learning (RL) can be interpreted as an inference or sampling problem, where the objective is to generate trajectories that maximize cumulative rewards. Let

$$\tau = (s_0, a_0, s_1, a_1, \ldots, a_T, s_{T+1})$$

denote a trajectory generated under transition dynamics $p(s_{t+1}|s_t, a_t)$ and initial-state distribution $p(s_0)$, with reward function $R_{\text{env}}(s_t, a_t)$.

Here, $s_t \in \mathcal{S}$ denotes the state at time $t$, $a_t \in \mathcal{A} = \mathbb{R}^N$ denotes the action at time $t$, and $T$ is the finite time horizon of the trajectory.

Instead of directly maximizing expected rewards, we define an *unnormalized target distribution* over action sequences $a_{0:T}$:

$$\widetilde{\pi}(a_{0:T}) = \int_{s_{0:T+1}} \prod_{t=0}^{T} p(s_{t+1}|s_t, a_t)\, \widetilde{\pi}(a_t|s_t)\, p(s_0)\, ds_{0:T+1},$$

with $\widetilde{\pi}(a_t|s_t) = \exp(\alpha\, R_{\text{env}}(s_t, a_t))$, $\alpha = \frac{1}{\mathcal{T}} > 0$, and where $\mathcal{T}$ is the temperature. This defines a *reward-weighted trajectory distribution*, where trajectories with higher cumulative rewards receive exponentially more probability mass. The normalized target distribution is

$$\pi(a_{0:T}) = \frac{\widetilde{\pi}(a_{0:T})}{Z}, \qquad Z = \int \widetilde{\pi}(a_{0:T})\, da_{0:T}.$$

A learned policy $q_\theta(a_{0:T})$ serves as a *variational approximation* to this target, defined by

$$q_\theta(a_{0:T}) = \int_{s_{0:T+1}} \prod_{t=0}^{T} p(s_{t+1}|s_t, a_t)\, q_\theta(a_t|s_t)\, p(s_0)\, ds_{0:T+1}.$$

Evaluating both $q_\theta(a_{0:T})$ and $\widetilde{\pi}(a_{0:T})$ is intractable due to integration over state states $s_t$ and thus directly minimizing any f-divergence (Csiszár, 1967) between $D_f(q_\theta(a_{0:T}) \,\|\, \pi(a_{0:T}))$ is not a valid option. To overcome this, a tractable upper bound on the f-divergence between trajectory distributions using the *data processing inequality* (van Erven & Harremoës, 2014) can be used:

$$D_f(q_\theta(a_{0:T}) \,\|\, \pi(a_{0:T})) \leq D_f(q_\theta(a_{0:T}, s_{0:T+1}) \,\|\, \pi(a_{0:T}, s_{0:T+1})). \tag{1}$$

where

$$q_\theta(a_{0:T}, s_{0:T+1}) = \prod_{t=0}^{T} p(s_{t+1}|s_t, a_t)\, q_\theta(a_t|s_t)\, p(s_0)$$

and $\quad \pi(a_{0:T}, s_{0:T+1}) = \prod_{t=0}^{T} p(s_{t+1}|s_t, a_t)\, \pi(a_t|s_t)\, p(s_0).$

Importantly, the data processing inequality ensures that the divergence between the joint state-action distributions provides an upper bound on the divergence defined over actions alone. Optimizing the upper bound, therefore, constrains the original objective. While it does not guarantee that a decrease in the upper bound leads to a strict decrease in the left-hand side, a reduction of the upper bound tightens the gap between the two divergences. The left-hand side is provably minimized only when the optimization drives the right-hand side down to the point where the inequality becomes tight. In this way, minimizing the right-hand side of Eq. 1 optimizes the variational policy $q_\theta(a_{0:T})$ to approximate the target distribution $\pi(a_{0:T})$, even though both cannot be evaluated in a tractable way.

## 2.1. Reinforcement Learning as Reverse Kullback–Leibler Divergence Minimization

Choosing $f = \mathrm{KL}$ yields the *reverse Kullback–Leibler divergence* objective which can be simplified to (see App. D.1):

$$D_{\mathrm{KL}}^{\mathcal{T}}\big(q_\theta(a_{0:T}, s_{0:T+1}) \,\big\|\, \pi(a_{0:T}, s_{0:T+1})\big)$$
$$\stackrel{C}{=} \sum_{t=0}^{T} \mathbb{E}_{s_t, a_t \sim q_\theta(a_t, s_t)}\big[\mathcal{T} \log q_\theta(a_t|s_t) - R_{\mathrm{env}}(s_t, a_t)\big], \quad (2)$$

where $C$ is a constant independent of $\theta$ and $q_\theta(a_t, s_t) = q_\theta(a_t|s_t) q_\theta(s_t)$. In the following, we will often write $s_t, a_t \sim q_\theta$, which means that first $s_t$ is sampled from $q_\theta(s_t)$ by unrolling the policy through the MDP and followed by sampling $a_t$ from $q_\theta(a_t|s_t)$.

Next we define $D_{\mathrm{KL}}^{\mathcal{T}} := \mathcal{T} D_{\mathrm{KL}}$ which is used as a loss in order to make the objective evaluable at $\mathcal{T} = 0$. In contrast to the original formulation, Eq. 2 is tractable and memory efficient gradient-based optimization is enabled by applying the *policy gradient theorem* (Sutton & Barto, 2018) (see App. D):

$$\nabla_\theta D_{\mathrm{KL}}^{\mathcal{T}}\big(q_\theta(a_{0:T}, s_{0:T+1}) \,\big\|\, \pi(a_{0:T}, s_{0:T+1})\big)$$
$$= \sum_{t=0}^{T} \mathbb{E}_{s_t, a_t \sim q_\theta}\Big[\big(\mathcal{T} \log q_\theta(a_t|s_t) - Q^{q_\theta}(s_t, a_t)\big) \nabla_\theta \log q_\theta(a_t|s_t)\Big], \quad (3)$$

where

$$Q^{q_\theta}(s_t, a_t) = R_{\mathrm{env}}(s_t, a_t) + \mathbb{E}_{s_{t+1} \sim p(\cdot|s_t, a_t)}\Big[V^{q_\theta}(s_{t+1})\Big],$$

$$V^{q_\theta}(s_t) = \delta_{t<T}\, \mathbb{E}_{a_t \sim q_\theta(a_t|s_t)}\big[Q^{q_\theta}(s_t, a_t) - \mathcal{T} \log q_\theta(a_t|s_t)\big],$$

and $(\delta_{t<T})$ zeros the value function at $(t = T)$ due to the finite-horizon setting.

By applying the *log-derivative trick in reverse*, the same problem can be expressed by using a surrogate loss that is more convenient to optimize in practice (see App. D.3):

$$\mathcal{L}_{\mathrm{ME}}(\theta) = \sum_{t=0}^{T} \mathbb{E}_{s_t \sim q_{\theta^*}} \Big[ D_{\mathrm{KL}}^{\mathcal{T}}\Big(q_\theta(a_t|s_t) \,\Big\|\, \frac{\exp(\alpha\, Q^{q_{\theta^*}}(s_t, a_t))}{Z(s_t)}\Big)\Big], \quad (4)$$

where the $*$ in $\theta^*$ denotes the application of a $\mathrm{stop\_grad}$ operation, which means that gradients are not propagated through these terms. In practice, the outer summation over time $t = 0, \ldots, T$ does not need to be evaluated exhaustively at every optimization step. Instead, it can be approximated via *Monte Carlo integration* by randomly sampling a subset of time steps. This yields an unbiased estimator of the full objective while avoiding backpropagation through all $T$ steps, thereby reducing computational cost. Concretely, the surrogate loss can be estimated with

$$\widehat{\mathcal{L}}_{\mathrm{ME}}(\theta) \approx \frac{T}{|\mathcal{U}|} \sum_{t \in \mathcal{U}} \mathbb{E}_{s_t \sim q_{\theta^*}} \Big[ \mathcal{L}_{\mathrm{ME}}(\theta, s_t) \Big]$$

where

$$\mathcal{L}_{\mathrm{ME}}(\theta, s_t) := \mathcal{T} D_{\mathrm{KL}}\Big(q_\theta(a_t|s_t) \,\Big\|\, \frac{\exp(\alpha\, Q^{q_{\theta^*}}(s_t, a_t))}{Z(s_t)}\Big)$$

and $\mathcal{U}$ denotes a minibatch of $|\mathcal{U}|$ uniformly sampled time indices.

This surrogate loss recovers the structure used as a basis in many different RL algorithms such as **Soft Actor-Critic (SAC)** (Haarnoja et al., 2018), REPPO (Voelcker et al., 2025), **Trust Region Policy Optimization (TRPO)** (Schulman et al., 2015), and **Proximal Policy Optimization (PPO)** (Schulman et al., 2017), and corresponds to the well-known *Maximum Entropy Reinforcement Learning* (ME-RL) objective, in which the expected reward is augmented with the policy entropy to encourage exploration and robustness.

It is important to note that the gradient in Eq. 4 is only exactly the same as Eq. 3 for the *first* policy update step, since the state distribution $q_\theta(s_t)$ in the expectation of Eq. 3 and the Q-function depend on the current policy parameters. After an update in Eq. 4 $\theta^*$ becomes $\theta_{\mathrm{old}}$, and thus the training samples are no longer strictly on-policy, and naively reusing it introduces bias, i.e. the gradient of Eq. 4 deviates from the gradient of the original objective in Eq. 3. On-policy algorithms such as TRPO and PPO therefore explicitly restrict the deviation between the current policy $\theta$ and the old policy $\theta^*$, which is used to collect the data. This is typically done by constraining or penalizing deviations between $q_\theta$ and $q_{\theta^*}$. These mechanisms ensure that the state distribution remains close enough for the gradient estimate to remain accurate. For a summary of some of these algorithms, see App. D.4.

## 2.2. Connection between the Log Variance Loss and Off-Policy Surrogate Objectives

In contrast, SAC is an off-policy algorithm, i.e. training samples are drawn from a replay buffer containing data

collected from older policies, for which the divergence to $q_\theta$ may be very large. Nevertheless, as shown in the next section, its actor update admits the following justification.

**Proposition 2.1** (Off-Policy ME-RL Surrogate via Log Variance Loss). *Let $\omega$ be an off-policy trajectory distribution that is absolutely continuous with respect to both $\pi$ and $q_\theta$. Consider the Log Variance (LV) loss, which is equivalent to the* Trajectory Balance *objective commonly used in GFlow Networks (Bengio et al., 2021b; Malkin et al., 2022b; Richter & Berner, 2024):*

$$D_{LV}^\omega(q_\theta(\tau)\|\pi(\tau)) \;=\; \frac{1}{2}\,\mathbb{E}_{\tau\sim\omega}\left[\left(\log\frac{q_\theta(\tau)}{\pi(\tau)} - b_\theta^\omega\right)^2\right], \tag{5}$$

*where $b_\theta^\omega = \mathbb{E}_{\tau\sim\omega}\big[\log\frac{q_\theta(\tau)}{\pi(\tau)}\big]$ and $\tau = (a_{0:T}, s_{0:T+1})$.*

*If states are sampled from an off-policy state distribution $\mathcal{B}$ (e.g. a replay buffer) and actions are sampled on-policy, $\omega(s_t, a_t) = \mathcal{B}(s_t)\,q_{\theta*}(a_t \mid s_t)$, then minimizing $D_{LV}^\omega$ gives rise to a policy-gradient-like update (see App. E) whose associated surrogate objective can be written as*

$$\mathcal{L}_{ME}^{\text{Off-Policy}}(\theta, s_t) = D_{\text{KL}}^{\mathcal{T}}\Big(q_\theta(a_t \mid s_t) \,\Big\|\, \frac{\exp(\alpha Q^{\omega, q_{\theta*}}(s_t, a_t))}{Z(s_t)}\Big), \tag{6}$$

*where importantly $s_t \sim \mathcal{B}(s_t)$,*

$$Q^{\omega, q_{\theta*}}(s_t, a_t) = R_{\text{env}}(s_t, a_t) + \mathbb{E}_{s_{t+1}}\big[V^{\omega, q_{\theta*}}(s_{t+1})\big],$$

*and*

$$V^{\omega, q_{\theta*}}(s_t) = \delta_{t<T}\,\mathbb{E}_{a_t\sim\omega(a_t|s_t)}\big[Q^{\omega, q_{\theta*}}(s_t, a_t) - \mathcal{T}\log q_{\theta*}(a_t \mid s_t)\big].$$

*Under these conditions, the resulting surrogate objective coincides with the KL-based actor loss used in Soft Actor-Critic.*

The full derivation of Proposition 2.1, is provided in App. E. The key implication is that the off-policy ME-RL surrogate objective in Eq. 4 is justified by the LV loss provided that actions in Eq. 6 are sampled on-policy. Two important caveats follow. First, the LV loss is not an $f$-divergence (Malkin et al., 2022b) and does not generally satisfy the data-processing inequality (Sanokowski et al., 2025b). Consequently, unlike the reverse-KL objective in Eq. 2, minimizing $D_{LV}^\omega$ does not admit a direct information-theoretic interpretation as an upper bound on a trajectory-level divergence. Second, on-policy methods can be interpreted within the same framework: the initial on-policy update coincides with the reverse-KL surrogate gradient in Eq. 3, while subsequent updates deviate because state samples come from $q_{\theta_{\text{old}}}$ rather than from $q_\theta$.

### 2.3. Maximum Entropy Wasserstein Policy Optimization

Starting from the reverse–KL surrogate objective in Eq. 4, this quantity can be interpreted as a functional over action distributions and therefore it is possible to derive Wasserstein gradient flow policy updates (Benamou & Brenier, 2000; Neklyudov et al., 2023). In contrast to the reward-only functional considered in (Pfau et al., 2025), the reverse–KL objective introduces the additional entropy-dependent term $\log q_\theta(a_t \mid s_t)$ inside the flow, leading to a slightly different velocity field and hence a modified parametric projection. We derive *Maximum Entropy WPO (ME-WPO)* as the Maximum Entropy Wasserstein gradient-flow analogue of the ME-RL by projecting the Wasserstein flow of Eq. 4 onto the parameter space. This yields the following surrogate loss (see App. F):

$$\mathcal{L}_{\text{ME-WPO}}(\theta, s_t) = \frac{\mathcal{T}}{2}\,\mathbb{E}_{a_t}\left[\left\|\nabla_{a_t}\Big(\log q_\theta(a_t|s_t) - \alpha Q^{q_{\theta*}}(s_t, a_t)\Big)\right\|^2\right]. \tag{7}$$

where $a_t \sim q_{\theta*}(a_t|s_t)$. Compared to the KL-based loss in Eq. 4, this loss matches the scores of the log probabilities instead of the log probabilities themselves. For $\mathcal{T} = 0$, the gradient of Eq. 7 with respect to $\theta$ coincides with the gradient of WPO. Importantly, the inverse Fisher information matrix $\mathcal{F}_{\theta\theta}^{-1}$ or its approximations should be applied after computing the gradient of Eq. 7 to perform the natural-gradient preconditioning. For practical implementation, we follow (Pfau et al., 2025) by approximating the Fisher matrix using the following gradient transformations derived from natural gradient updates for the mean and standard deviation of Gaussian distributions: $\nabla_\mu \to \sigma^2\nabla_\mu$ and $\nabla_\sigma \to \frac{1}{2}\sigma^2\nabla_\sigma$, yielding a simple preconditioner that can easily be implemented.

### 2.4. Diffusion Samplers

We now introduce the diffusion process used to model the policy distribution $q_\theta(a^{(0)})$ with the aim of approximating an intractable target distribution $\pi(a^{(0)})$. For clarity, we drop the state-dependence and assume diagonal Gaussian noise.

**Continuous-Time Formulation:** Diffusion samplers define a stochastic differential equation (SDE) that gradually perturbs a target distribution into a stationary distribution, along with a corresponding reverse process that aims to reconstruct it. For the variance-preserving (VP) SDE (Song et al., 2021b), the forward process is

$$da^{(k)} = -\tfrac{1}{2}\beta(k)\,a^{(k)}\,dk + \nu\sqrt{\beta(k)}\,dW_k, \quad k \in [0, K],$$

where $\beta(k) \in \mathbb{R}_+^N$ is a noise schedule and $W_k$ a standard Wiener process. Starting from $k = 0$, this forward process diffuses the target distribution into a simple Gaussian prior $q(a^{(K)}) = \mathcal{N}(0, \nu^2)$ at $k = K$. The corresponding reverse-time SDE is

$$da^{(k)} = \Big[-\tfrac{1}{2}\beta(k)\,a^{(k)} - \nu^2\beta(k)\,\nabla_{a^{(k)}}\log\pi_k(a^{(k)})\Big]dk + \nu\sqrt{\beta(k)}\,d\bar{W}_k,$$

where $\bar{W}_s$ denotes a reverse Wiener process and $\log\pi_k(a^{(k)}) = \log\big(\int\pi(a^{(k)}|a^{(0)})\pi(a^{(0)})da^{(0)}\big)$ which is intractable to compute.

**Discrete-Time Approximation:** Applying Euler–Maruyama integration with step size $\Delta_k$ (from $k = K$ to $k = 1$ in reverse direction) and by defining $\delta_k := \beta_k \Delta_k$ yield the following discrete update rules. The **forward diffusion** step from $a^{(k-1)}$ to $a^{(k)}$ is

$$a^{(k)} = (1 - \tfrac{1}{2}\delta_{k-1})\,a^{(k-1)} + \nu\sqrt{\delta_{k-1}}\,\varepsilon_{k-1}, \qquad \varepsilon_{k-1} \sim \mathcal{N}(0, I),$$

so that the forward diffusion kernel is given by:

$$\pi(a^{(k)}|a^{(k-1)}) = \mathcal{N}\Big((1 - \tfrac{1}{2}\delta_{k-1})\,a^{(k-1)}, \delta_{k-1}\,\nu^2\Big).$$

The **reverse diffusion** step, parameterized by $\theta$, is

$$a^{(k-1)} = (1 + \tfrac{1}{2}\delta_k)\,a^{(k)} + \nu^2\,\delta_k\,s_\theta(a^{(k)}, k) + \nu\sqrt{\delta_k}\,\varepsilon_k,$$

with the reverse diffusion kernel

$$q_\theta(a^{(k-1)}|a^{(k)}) = \mathcal{N}\Big((1 + \tfrac{1}{2}\delta_k)\,a^{(k)} + \nu^2\,\delta_k\,s_\theta(a^{(k)}, k), \delta_k\,\nu^2\Big),$$

where $s_\theta(a^{(k)}, k)$ is a neural network-based approximation of the intractable score function $\nabla_{a^{(k)}} \log \pi_k(a^{(k)})$.

**Intractability of the Marginal:** The marginal of the learned model is obtained by integrating all intermediate diffusion variables:

$$q_\theta(a^{(0)}) = \int \prod_{k=K}^{1} q_\theta(a^{(k-1)} \mid a^{(k)})\,q(a^{(K)})\,da^{(1:K)}.$$

This integral is, in general, intractable, and consequently, the reverse-KL divergence $D_{\mathrm{KL}}(q_\theta(a^{(0)}) \,\|\, \pi(a^{(0)}))$ cannot be directly used as a loss function. However, by the *data processing inequality*, the KL divergence between the marginal distributions can be upper bounded by the divergence between the joint distributions in the following way:

$$D_{\mathrm{KL}}\big(q_\theta(a^{(0)}) \,\|\, \pi(a^{(0)})\big) \leq D_{\mathrm{KL}}\big(q_\theta(a^{(0:K)}) \,\|\, \pi_\theta(a^{(0:K)})\big),$$

which is tractable as

$$q_\theta(a^{(0:K)}) = \prod_{k=K}^{1} q_\theta(a^{(k-1)} \mid a^{(k)})\,q(a^{(K)})$$

$$\text{and} \quad \pi_\theta(a^{(0:K)}) = \prod_{k=1}^{K} \pi_\theta(a^{(k)} \mid a^{(k-1)})\,\pi(a^{(0)}).$$

As explained in Sec. 2, by optimizing the right-hand side, we implicitly optimize the left-hand side of the equation.

**Learnable Diffusion Coefficients.** In contrast to fixed noise schedules, we follow (Celik et al., 2025) consider the setting where the diffusion coefficients are *learnable*. Specifically, the noise schedule $\beta(k)$ (and consequently its discretized counterpart $\beta_k$) is parameterized and optimized jointly with the score network. As a result, the forward and reverse diffusion processes share learnable parameters through the diffusion coefficients. This coupling allows the model to adapt the amount of injected

noise to the target distribution, yielding a more flexible sampler. We therefore introduce the augmented parameter vector $\theta = (\theta', \{\beta_K, \ldots, \beta_0\})$, where $\theta'$ are redefined to the parameters of the score network and write the resulting forward diffusion kernel as $\pi_\theta$. Importantly, the forward diffusion kernel $\pi_\theta(a^{(k)} \mid a^{(k-1)})$ depends only on the diffusion coefficients $\{\beta_K, \ldots, \beta_0\}$ and does not depend on the score network parameters $\theta'$.

## 3. Method

When using diffusion policies in RL, optimizing the right-hand side of Eq. 1 is intractable as the marginal $q_\theta(a_t \mid s_t)$ cannot be easily evaluated. Therefore, similarly to Sec. 2.4, we apply the data-processing inequality once again to Eq. 1 by including the joint probability of the whole reverse and forward diffusion processes over $a_t^{0:K}$. This yields

$$D_{\mathrm{KL}}\big(q_\theta(a_{0:T}, s_{0:T+1}) \,\|\, \pi(a_{0:T}, s_{0:T+1})\big)$$
$$\leq D_{\mathrm{KL}}\big(q_\theta(a_{0:T}^{0:K}, s_{0:T+1}) \,\|\, \pi_\theta(a_{0:T}^{0:K}, s_{0:T+1})\big), \tag{8}$$

where $q_\theta(a_t^{0:K}|s_t)$ and $\pi_\theta(a_t^{0:K}|s_t)$ denote the joint distributions

$$q_\theta(a_{0:T}^{0:K}, s_{0:T+1}) = \prod_{t=0}^{T} p(s_{t+1} \mid s_t, a_t^0)\,q_\theta(a_t^{0:K} \mid s_t)\,p(s_0)$$

$$\text{and} \quad \pi_\theta(a_{0:T}^{0:K}, s_{0:T+1}) = \prod_{t=0}^{T} p(s_{t+1} \mid s_t, a_t^0)\,\pi_\theta(a_t^{0:K} \mid s_t)\,p(s_0). \tag{9}$$

**KL decomposition:** The KL divergence between the joint diffusion policy $q_\theta$ and the reference trajectory distribution $\pi$ decomposes over time $t$ and diffusion index $k$ as

$$D_{\mathrm{KL}}^{\mathcal{T}}\Big(q_\theta(a_{0:T}^{0:K}, s_{0:T+1}) \,\|\, \pi_\theta(a_{0:T}^{0:K}, s_{0:T+1})\Big)$$
$$\overset{C}{=} \sum_{t=0, k=K}^{T, 1} \mathbb{E}_{q_\theta}\left[\mathcal{T}\log\frac{q_\theta(a_t^{k-1}|a_t^k, s_t)}{\pi_\theta(a_t^k|a_t^{k-1}, s_t)} - R_{\mathrm{env}}(s_t, a_t^0)\,\mathbf{1}_{\{k=1\}}\right], \tag{10}$$

where the expectation goes over $s_t, a_t^{k-1}, a_t^k \sim q_\theta$, $\mathbf{1}_{\{k=1\}}$ is the indicator function and $C$ collects all terms independent of $\theta$ and actions.

We prove in App. G that when taking the gradient of Eq. 10 with respect to $\theta$, the policy gradient theorem still holds and thus the following surrogate loss for *Diffusion-based Maximum Entropy Reinforcement Learning* can be used:

$$\widehat{\mathcal{L}}_{\mathrm{DME}}(\theta) \approx \frac{T\,K}{|\mathcal{U}_{T,K}|} \sum_{t,k \sim \mathcal{U}_{T,K}} \mathbb{E}_{\tilde{s}_{\tilde{t}} \sim q_{\theta*}}\Big[\mathcal{L}_{\mathrm{DME}}(\theta, \tilde{s}_{\tilde{t}})\Big]$$

with

$$\mathcal{L}_{\mathrm{DME}}(\theta, \tilde{s}_{\tilde{t}}) = D_{\mathrm{KL}}^{\mathcal{T}}\left(q_\theta(\cdot|\tilde{s}_{\tilde{t}}) \,\Big\|\, \pi_\theta(a_t^k|\cdot, s_t)\frac{\exp\big(\alpha\,Q_{\mathrm{DME}}^{q_{\theta*}}(\tilde{s}_{\tilde{t}}, \cdot)\big)}{Z(\tilde{s}_{\tilde{t}})}\right). \tag{11}$$

where the state time step $t$ and diffusion time step $k$ are sampled from the uniform distribution $\mathcal{U}_{T,K}$. The corresponding Q- and value functions are given by

$$Q_{\mathrm{DME}}^{q_{\theta*}}(\tilde{s}_{\tilde{t}}, a_t^{k-1}) = \tilde{R}_{\mathrm{DME}}(a_t^{k-1}, \tilde{s}_{\tilde{t}}) + \mathbb{E}_{\tilde{s}_{\tilde{t}+1}}\Big[V_{\mathrm{DME}}^{q_{\theta*}}(\tilde{s}_{\tilde{t}+1})\Big],$$

$$V_{\text{DME}}^{q_{\theta^*}}(\tilde{s}_{\tilde{t}}) = \delta_{t < T} \, \mathbb{E}_{a_t^{k-1}} \left[ Q_{\text{DME}}^{q_{\theta^*}}(\tilde{s}_{\tilde{t}}, a_t^{k-1}) - \mathcal{T} \log \frac{q_{\theta^*}(a_t^{k-1}|a_t^k, s_t)}{\pi_{\theta^*}(a_t^k|a_t^{k-1}, s_t)} \right],$$
$$\tag{12}$$

where $a_t^{k-1} \sim q_{\theta^*}(\cdot|\tilde{s})$ and $\tilde{s}_{\tilde{t}} = \tilde{s}_{\tilde{t}(t,k)} = (s_t, a_t^k, k)$ and $\tilde{R}_{\text{DME}}$ is defined in Eq. 13 (see Sec. 3.1). Importantly, compared to ME-RL, the value function in DME-RL includes the log ratio between forward and reverse diffusion transition probabilities.

## 3.1. Diffusion-Based Maximum Entropy Reinforcement Learning Markov Decision Process

We can formulate the resulting Diffusion-based Maximum Entropy MDP by flattening the original time steps ($t = 0, \ldots, T$) and the reverse diffusion steps ($k = K, \ldots, 1$) into a single augmented time index ($\tilde{t}$):

$$\tilde{t}(t, k) = t \, K + (K - k), \qquad \tilde{s}_{\tilde{t}(t,k)} = (s_t, a_t^k, k), \qquad \tilde{a}_{\tilde{t}(t,k)} = a_t^{k-1}.$$

The modified reward $\tilde{R}_{\text{DME}}$ is defined such that only at the last diffusion step ($k = 1$) the environment reward is called:

$$\tilde{R}_{\text{DME}}(\tilde{s}_{\tilde{t}(t,k)}, \tilde{a}_{\tilde{t}(t,k)}) = \begin{cases} 0, & k > 1, \\ R_{\text{env}}(s_t, a_t^0), & k = 1. \end{cases} \tag{13}$$

The augmented MDP transition kernel is

$$p(\tilde{s}_{\tilde{t}+1} \mid \tilde{s}_{\tilde{t}}, \tilde{a}_{\tilde{t}}) = \begin{cases} \delta\left(\tilde{s}_{\tilde{t}+1} = (s_t, a_t^{k-1}, k-1)\right), & k > 1, \\ p(s_{t+1} \mid s_t, a_t^0) \otimes q_{\text{prior}}(a_{t+1}^K) \otimes \delta(k - K), & k = 1 \end{cases}$$

This formulation explicitly captures the reverse diffusion steps as intermediate MDP states. For ($k > 1$), the MDP moves to the next diffusion step ($k - 1$) while keeping the environment state ($s_t$) fixed. When ($k = 1$), the environment transitions forward to ($s_{t+1}$) using the final actions ($a_t^0$) and the next diffusion chain starts with ($a_{t+1}^K$) sampled from the prior, effectively resetting the diffusion step index to ($K$). This ensures that the augmented MDP correctly integrates both the environment dynamics and the diffusion-based policy structure. This MDP is the same as in DPPO (Ren et al., 2025), however, unlike in DPPO the definition of the value function is different (see Eq. 12). Thus, our formulation matches the DPPO formulation at $\mathcal{T} = 0$ and is otherwise a generalization.

## 3.2. Diffusion-based Maximum Entropy Algorithms

**DME-PPO and DME-REPPO:** In this paper, we evaluate our framework by modifying PPO, REPPO and Maximum Entropy WPO to their Diffusion-based variations DME-PPO, DME-REPPO and DME-WPO. For DME-PPO we use the log-derivative trick combined with importance weights to optimize Eq. 11 and extend the clipping mechanism to learnable forward diffusion processes (see App. G.4), while for DME-REPPO we optimize Eq. 11 using the parametrization trick and adapt REPPOs auxiliary loss (Jaderberg et al., 2016) to diffusion-based MDPs (see

App. H.4 for more details). In the next section, we describe the DME-WPO loss. In practice, DME-WPO follows the same setup as DME-REPPO, but with the policy loss replaced by the DME-WPO loss.

**Diffusion Wasserstein Policy Optimization:** We extend diffusion-based policies to *Wasserstein Policy Optimization (WPO)* (Pfau et al., 2025), which can be derived by projecting the Wasserstein Gradient Flow of the surrogate loss in Eq. 11 via the reverse-KL with respect to the reverse diffusion kernel and forward-KL with respect to the forward diffusion kernel into parameter space (see App. G.5). Thus we obtain the following surrogate objective for **DME-WPO**:

$$\mathcal{L}_{\text{DME-WPO}}(\theta, \tilde{s}_{\tilde{t}})$$
$$= \frac{\mathcal{T}}{2} \mathbb{E}_{a_t^{k-1}} \left[ \left|\left| \nabla_{a_t^{k-1}} \left( \log \frac{q_\theta(a_t^{k-1}|a_t^k, s_t)}{\pi_\theta(a_t^k|a_t^{k-1}, s_t)} - \alpha Q_{\text{DME}}^{q_{\theta^*}}(\tilde{s}_{\tilde{t}}, a_t^{k-1}) \right) \right|\right|^2 \right],$$
$$\tag{14}$$

where $a_t^{k-1} \sim q_{\theta^*}(\cdot|\tilde{s}_{\tilde{t}})$. In projected Wasserstein Gradient Flows, the resulting gradients are typically preconditioned by the inverse Fisher matrix. In our diffusion setting, however, the reverse diffusion kernel for VP-SDEs is not a simple Gaussian, so the natural-gradient updates from Pfau et al. (2025) cannot be applied. However, by using Lagrange multipliers to constrain the deviation between $q_\theta$ and $q_{\theta^*}$ an implicit preconditioning effect is already provided, and while we experimented with diagonal Fisher approximations, they were unstable on some tasks. For DME-WPO, we therefore omit explicit preconditioning in all experiments.

## 4. Experiments

In this section, we evaluate the proposed diffusion-augmented RL algorithms (**DME-PPO**, **DME-REPPO**, and **DME-WPO**) on a suite of standard continuous-control benchmarks. We compare them against REPPO (Voelcker et al., 2025) and PPO (Schulman et al., 2017), using the results reported in (Voelcker et al., 2025) (which we denote as PPO (r) in our experiments). We also include comparisons to a concurrent method that combines DIME (Celik et al., 2025) with REPPO ("REPPO-DIME"); we will cite the corresponding paper once it becomes publicly available. We evaluate all diffusion-based algorithms (including REPPO-DIME) using 8 diffusion steps (see App. H for implementation details and a more detailed explanation of hyperparameter settings). We additionally compare WPO (Pfau et al., 2025) to its Maximum Entropy generalized variation ME-WPO. The methods are evaluated on various standard control tasks from the DeepMind Control suite (Zakka et al., 2025) (see Fig. 1), where we are considering the massively parallel on-policy RL setting (Makoviychuk et al., 2021; Zakka et al., 2025; Tao et al., 2025). In these experiments, we run 7 independent seeds for all algorithms. Our code is based on the code base of REPPO and for each method, the hyperparameters across all environments are

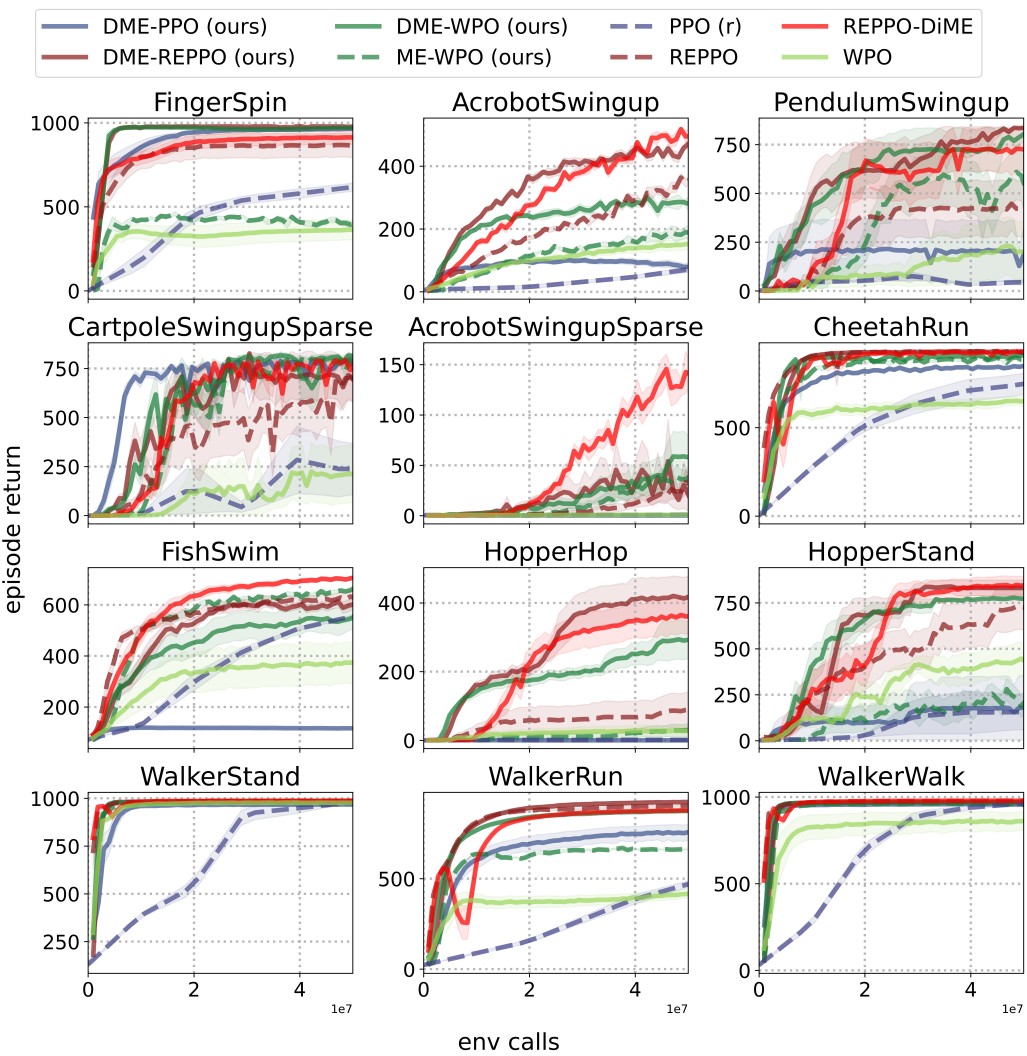

*Figure 1.* Performance comparison of DME-RL-based algorithms to their vanilla counterparts and to REPPO-DIME. Results are averaged across five independent seeds, along with their standard errors.

kept the same. Experiments on the humanoid walk, run, and standup benchmarks are excluded from this evaluation due to frequent numerical instability issues (NaN values) in the corresponding MuJoCo playground (Zakka et al., 2025) environments.

### 4.1. Main Results

We first validate the ME-WPO generalization by comparing WPO and ME-WPO across a wide range of DeepMind Control tasks. Our ME-WPO and WPO implementations are built on top of REPPO by replacing the policy loss with the Wasserstein Policy Gradient losses. Thus, ME-WPO uses automatic temperature tuning using Lagrange multipliers (see App. D.5), and for WPO, we set the temperature to zero. Results in Fig. 1 and Fig. 2 show that ME-WPO achieves better and more stable returns than WPO, though REPPO still

outperforms ME-WPO. Next, we compare PPO with DME-PPO, and DME-WPO/DME-REPPO with their vanilla counterparts (ME-WPO/REPPO), and include REPPO-DIME as a diffusion baseline. Across tasks, diffusion-based methods (REPPO-DIME, DME-REPPO, and DME-WPO) tend to outperform non-diffusion algorithms, particularly on harder tasks such as **HopperHop**, **HopperStand**, **FingerSpin**, **AcrobotSwingup**, **PendulumSwingup**, and **CartpoleSwingupSparse**. On **ArcrobotSwingupSparse**, we observe that DME-REPPO and DME-WPO perform much worse and less stably across different seeds than REPPO-DIME. Although we have adapted the auxiliary loss, used in REPPO, to diffusion-based MDPs (see App. H.4), this choice does not appear optimal and does not work reliably across different seeds. Improving this is an interesting direction for future work.

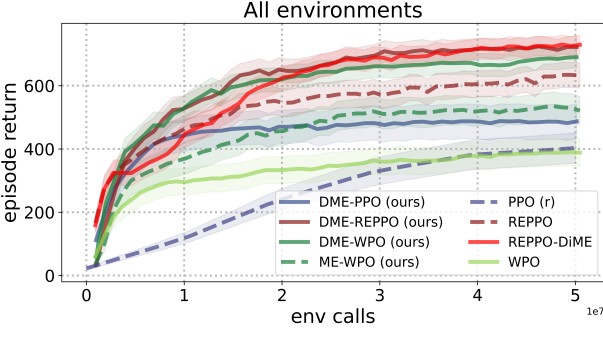

*Figure 2.* Average results across all environments considered in Fig. 1

We also observe different learning dynamics: DME-WPO and DME-REPPO typically improve more slowly early on, likely because the Q-function must first learn the diffusion dynamics. DME-PPO learns much faster than PPO and converges to a better final performance. Overall, DME-RL variants consistently outperform their base algorithms, and DME-WPO/DME-REPPO perform comparably to REPPO-DIME while offering greater memory flexibility during training. The observed improvements of REPPO-DIME, DME-REPPO, compared to REPPO on the averaged results across all environments are statistically significant.

Finally, we report the average runtime across all environments for each algorithm in App. B (Table 1). The results show that WPO and ME-WPO require approximately $1.3\times$ more runtime than REPPO, likely due to the additional computation of the Q-function scores. REPPO-DIME, in comparison, requires about $3.3\times$ more runtime, while DME-PPO, DME-REPPO, and DME-WPO demand approximately $3.6\times$, $4.4\times$, and $5.5\times$ more runtime, respectively, compared to REPPO. The increased training time for DME-based algorithms, relative to REPPO-DIME, stems from our experimental design. To demonstrate that our algorithm does not require backpropagation through all diffusion steps, we chose a configuration with twice as many gradient update steps by using a smaller batch size, which inherently extends the overall training time.

## 5. Conclusion and Future Work

In this work, we introduced a principled framework for generalizing reinforcement learning to diffusion-based policies, which we call **Diffusion-based Maximum Entropy Reinforcement Learning (DME-RL)**. This framework, derived from reverse KL minimization, generalizes prior diffusion-based RL methodologies and shows promising results across a wide range of popular control benchmarks. We further make key theoretical findings: first, establishing a direct equivalence by showing that the off-policy Maximum Entropy RL surrogate loss (e.g., used in SAC) can be derived

from the log-variance (trajectory balance) loss (Richter et al., 2020; Malkin et al., 2022a); and second, deriving a Maximum Entropy formulation of the recently proposed Wasserstein Policy Optimization (WPO) (Pfau et al., 2025).

Building on our DME-RL framework, we proposed three novel diffusion-augmented algorithms: **DME-PPO**, **DME-REPPO**, and **DME-WPO**. These methods are practical and easily implementable, requiring only minor modifications to their respective algorithm. We demonstrate that they achieve strong performance on challenging continuous-control benchmarks, improving the final average returns compared to their respective base algorithms (REPPO, ME-WPO and PPO). Looking ahead, our framework offers a promising mechanism for fine-tuning large diffusion-based policies or vision–language–action (VLA) models without requiring backpropagation through all diffusion steps. A key limitation of the current formulation is that training requires iterating through the $Q$-function or value function at every diffusion step. This overhead could be reduced through specialized architectures, for example, by encoding the state and the previous diffusion action step using separate encoders. In such a design, the state encoder could be reused across diffusion steps and only recomputed once the environment state changes after the final diffusion step. The procedure can also be applied to the policy network.

There are several promising directions for future work, such as making the diffusion prior distribution learnable and state-dependent, allowing both the mean and variance to depend on the current state, and explicitly incorporating this dependency into the underlying Diffusion MDP formulation. Beyond these directions, our framework could be extended to use *diffusion bridge samplers* for diffusion policy parametrization (Vargas et al., 2024; Richter & Berner, 2024; Sanokowski et al., 2025b) to enable more efficient trajectory-level modeling. In principle, the framework can be applied to discrete action spaces and is particularly interesting in settings with extremely large action spaces where direct sampling is infeasible. Examples of such problems are combinatorial optimization problems explored in NeuralCO (Karalias & Loukas, 2020). Notably, our approach is more general than some existing methods, such as (Sanokowski et al., 2025a), as it naturally accommodates probabilistic transition dynamics. Finally, diffusion-based RL in discrete domains opens up applications to reinforcement learning from human feedback (RLHF) (Ouyang et al., 2022) for Diffusion Language Models (Nie et al., 2025), enabling memory-efficient RLHF-based fine-tuning.

## Impact Statement

Diffusion-based reinforcement learning has the potential to improve decision-making systems in real-world applications where sample efficiency and exploration are critical. By en-

abling policies that can model complex, multimodal action distributions, our framework could lead to more adaptive and reliable robotics, autonomous systems, and other AI agents that interact with dynamic environments.

At the same time, as with all powerful generative and decision-making models, there are societal risks. More capable RL agents could be misused in autonomous systems for harmful purposes. Careful evaluation, robust safety constraints, and ethical deployment practices are essential to ensure these technologies benefit society responsibly.

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

## A. Related Work

### A.1. Sampling from Unnormalized Target Distributions

Sampling from unnormalized target distributions is a vibrant and active research area. Classical methods, such as Markov Chain Monte Carlo (MCMC) (Metropolis et al., 1953), have long been foundational, but recent advances increasingly leverage neural networks to approximate target distributions (Noé & Wu, 2018; Wu et al., 2019). These approaches find broad applications, including molecule configuration prediction (Noé & Wu, 2018), statistical physics (Wu et al., 2019), Monte Carlo integration (Müller et al., 2019), and combinatorial optimization (Hibat-Allah et al., 2021). Diffusion samplers offer an expressive solution by transporting noise samples toward high-density regions of the target distribution $\pi$ via a learned reverse diffusion process. These methods have been successfully applied in both continuous (Zhang & Chen, 2022; Berner et al., 2022; Vargas et al., 2023; 2024; Richter & Berner, 2024) and discrete domains (Sanokowski et al., 2024; 2025a), demonstrating robust performance on challenging sampling problems. Adjoint Matching (Domingo-Enrich et al., 2025) also employs a score-matching-based loss similar to our DME-WPO loss, though it is derived from the adjoint method and, in contrast to our method, does not allow for learnable diffusion coefficients. Physics-informed neural network (PINN) (Raissi et al., 2019) losses, derived from the continuity equation of continuous normalizing flows, have been explored in (Tian et al., 2024; Máté & Fleuret, 2023; Fan et al., 2024). Reinforcement learning (RL) naturally aligns with the unnormalized-sampling paradigm. A substantial body of work has established that RL can be formulated as probabilistic inference (Todorov, 2008; Ziebart, 2010; Kappen et al., 2012; Levine, 2018; Abdolmaleki et al., 2018).

### A.2. Diffusion Models in Reinforcement Learning

Diffusion models have recently gained traction in RL for representing complex action distributions. Many works optimize forward-KL objectives (Dong et al., 2025; Ma et al., 2025) by replacing the reverse KL divergence in Eq. 4 with a forward KL divergence. However, forward KL requires samples from the target distribution, which are typically unavailable. Neural importance sampling is often used to mitigate this challenge, but it introduces bias, high variance, and *mode-covering* behavior—an undesirable trait in RL, where suboptimal actions should be avoided entirely. DIME (Celik et al., 2025) integrates diffusion models with Soft Actor-Critic (SAC), but unlike our approach, it does not evaluate a Q-function at each diffusion step. Instead, its objective is based on applying the Data Processing Inequality directly to the surrogate loss in Eq. 4. This method requires backpropagation through the entire diffusion chain for every environment action, which becomes memory-intensive for large numbers of diffusion steps $T$. In contrast, our diffusion-based RL approaches train the diffusion model itself via RL, as in (Sanokowski et al., 2025a). This allows flexible memory usage by reducing the minibatch size, since the loss decomposes over diffusion steps and can be estimated from small subsets without backpropagating through the full chain. DPPO (Ren et al., 2025) corresponds to a special case of DME-PPO when $\tau = 0$. Our DME-PPO formulation generalizes this method to arbitrary temperatures. Recent works have also explored flow models in RL (Liu et al., 2025; Park et al., 2025; Zhang et al., 2025; McAllister et al., 2025), further expanding the intersection of generative models and RL.

## B. Runtime Comparison

In the following, we present a runtime comparison for each method averaged across all environments. Each method is run on a single A100 NVIDIA GPU.

| Method | Runtime (hours) | Factor vs REPPO | Env count |
|---|---|---|---|
| REPPO | 0.47 | 1.00 | 12 |
| ME-WPO (ours) | 0.64 | 1.32 | 12 |
| WPO | 0.64 | 1.32 | 12 |
| REPPO-DIME | 1.49 | 3.33 | 12 |
| DME-PPO (ours) | 1.69 | 3.60 | 12 |
| DME-REPPO (ours) | 2.08 | 4.44 | 12 |
| DME-WPO (ours) | 2.61 | 5.47 | 12 |

*Table 1.* Runtime Comparison of each method averaged over all environments.

## C. Further Directions for Future Work

Another key consideration is optimization stability, particularly when using the HL-Gauss loss (Farebrother et al., 2024) in DME-REPPO or DME-WPO with automatic temperature tuning. If the temperature overshoots, the log-ratios, which are scaled by the temperature, can grow excessively large and unbounded. This can push $Q$-targets outside the fixed binning range of the HL loss, resulting in numerical instability. While reducing the learning rate for the Lagrange multipliers can partially mitigate this issue, we see this as an important area for future research. We also found that the minibatch size significantly influences the learned temperature and KL regularization schedule and can make learning unstable, primarily due to noisy updates in the Lagrange multipliers. Smoothing techniques could potentially enhance robustness in this context. Furthermore, in Eq. 11, the diffusion time step is currently sampled uniformly. An alternative approach could involve sampling later diffusion steps more frequently using importance sampling, a technique commonly employed in data-based diffusion models (Song et al., 2021a), which may further improve learning efficiency.

## D. Derivation of the Surrogate KL Objective

In this appendix, we provide the complete derivation of the surrogate Maximum Entropy Objective from Eq. 4 starting from the trajectory-level KL divergence

$$D_{\mathrm{KL}}\big(q_\theta(a_{0:T}, s_{0:T+1}) \,\big\|\, \pi(a_{0:T}, s_{0:T+1})\big)$$

to the per-state surrogate KL objective used in maximum-entropy reinforcement learning. We first show that the environment dynamics cancel within the log ratios between $q_\theta$ and $\pi$, then explain why the policy gradient theorem applies, and finally show how the reverse log-derivative trick leads to a tractable surrogate loss.

### D.1. Trajectory Distributions and Cancellation of Dynamics

Recall the trajectory distributions:

$$q_\theta(a_{0:T}, s_{0:T+1}) = \prod_{t=0}^{T} p(s_{t+1} \mid s_t, a_t)\, q_\theta(a_t \mid s_t)\, p(s_0),$$

$$\pi(a_{0:T}, s_{0:T+1}) = \prod_{t=0}^{T} p(s_{t+1} \mid s_t, a_t)\, \pi(a_t \mid s_t)\, p(s_0).$$

Both distributions contain the same environment dynamics and initial state distribution. Inside log probability ratios $\log \frac{q_\theta(a_{0:T}, s_{0:T})}{\pi(a_{0:T}, s_{0:T})}$ within the KL divergence we have,

$$\log p(s_{t+1} \mid s_t, a_t) - \log p(s_{t+1} \mid s_t, a_t) = 0, \qquad \log p(s_0) - \log p(s_0) = 0,$$

so these terms cancel exactly. Thus the trajectory-level KL reduces to a sum of per-timestep policy KL terms:

$$\log q_\theta(a_{0:T}, s_{0:T}) - \log \pi(a_{0:T}, s_{0:T}) = \sum_{t=0}^{T} \big( \log q_\theta(a_t \mid s_t) - \log \pi(a_t \mid s_t) \big).$$

with $\log \pi(a_t \mid s_t) = R_{\mathrm{env}}(a_t, s_t) + C_t$ leading with $C := \sum_{t=0}^{T} C_t$ to

$$D_{\mathrm{KL}}^{\mathcal{T}}\big(q_\theta(a_{0:T}, s_{0:T}) \,\big\|\, \pi(a_{0:T}, s_{0:T})\big) = \sum_{t=0}^{T} \mathbb{E}_{s_t, a_t \sim q_\theta(a_t, s_t)} \big[ \mathcal{T} \log q_\theta(a_t \mid s_t) - R_{\mathrm{env}}(s_t, a_t) \big] + C.$$

### D.2. Why the Policy Gradient Theorem Applies

We begin with the reverse–KL objective

$$D_{\mathrm{KL}}(q_\theta(a_{0:T}, s_{0:T}) \,\|\, \pi(a_{0:T}, s_{0:T})) = \mathbb{E}_{(a_{0:T}, s_{0:T}) \sim q_\theta} \left[ \sum_{t=0}^{T} \big( \log q_\theta(a_t \mid s_t) - \log \pi(a_t \mid s_t) \big) \right].$$

where $\log \pi(a_t \mid s_t) = \alpha R_{\mathrm{env}}(s_t, a_t)$. To express this in the standard RL framework, we rewrite it as the expectation of a sum of *modified rewards*. Define

$$\tilde{r}(s_t, a_t) := R_{\mathrm{env}}(s_t, a_t) - \mathcal{T} \log q_\theta(a_t \mid s_t), \qquad \tilde{R}(\tau) = \sum_{t=0}^{T} \tilde{r}(s_t, a_t).$$

The key identity

$$\mathbb{E}_{a_t \sim q_\theta}[\nabla_\theta \log q_\theta(a_t \mid s_t)] = 0$$

implies that terms of the form $\nabla_\theta\big(f(a_t, s_t) \log q_\theta(a_t \mid s_t)\big)$ behave, for gradient purposes, exactly as if $f(a_t, s_t) \log q_\theta(a_t \mid s_t)$ were independent of $\theta$. Thus, the reward dependence of $-\mathcal{T} \log q_\theta(a_t \mid s_t)$ on $\theta$ does not introduce any additional terms in the gradient, and we can treat $\tilde{r}(s_t, a_t)$ as a valid reward function for the purpose of applying the policy gradient theorem.

Hence, the KL-based objective is equivalent (up to a sign) to the standard RL objective

$$J(\theta) = \mathbb{E}_{\tau \sim q_\theta}\left[\sum_{t=0}^{T} \tilde{r}(s_t, a_t)\right], \qquad \text{with} \quad J(\theta) = -\mathcal{T} D_{\mathrm{KL}}(q_\theta(a_{0:T}, s_{0:T}) \,\|\, \pi(a_{0:T}, s_{0:T})).$$

Since $J(\theta)$ is now the expected cumulative reward under policy $q_\theta$, the policy gradient theorem (Sutton & Barto, 2018) applies directly:

$$\nabla_\theta D_{\mathrm{KL}}^{\mathcal{T}}(q_\theta(a_{0:T}, s_{0:T}) \| \pi(a_{0:T}, s_{0:T})) = -\sum_{t=0}^{T} \mathbb{E}_{s_t, a_t \sim q_\theta}\left[\left(\tilde{r}(s_t, a_t) + \mathbb{E}_{s_{t+1}}\left[V^{q_\theta}(s_{t+1})\right]\right) \nabla_\theta \log q_\theta(a_t \mid s_t)\right]$$

$$= \sum_{t=0}^{T} \mathbb{E}_{s_t, a_t \sim q_\theta}\left[\left(\mathcal{T} \log q_\theta(a_t \mid s_t) - R_{\mathrm{env}}(s_t, a_t) - \mathbb{E}_{s_{t+1}}\left[V^{q_\theta}(s_{t+1})\right]\right) \nabla_\theta \log q_\theta(a_t \mid s_t)\right]$$

Substituting

$$Q^{q_\theta}(s_t, a_t) = R_{\mathrm{env}}(s_t, a_t) + \mathbb{E}_{s_{t+1} \sim p(\cdot \mid s_t, a_t)}\left[V^{q_\theta}(s_{t+1})\right]$$

and

$$V^{q_\theta}(s_t) = \delta_{t<T} \, \mathbb{E}_{a_t \sim q_\theta(a_t \mid s_t)}\left[Q^{q_\theta}(s_t, a_t) - \mathcal{T} \log q_\theta(a_t \mid s_t)\right]$$

and $\tilde{r}(s_t, a_t) = R_{\mathrm{env}}(s_t, a_t) - \mathcal{T} \log q_\theta(a_t \mid s_t)$ yields

$$\nabla_\theta D_{\mathrm{KL}}^{\mathcal{T}}(q_\theta(a_{0:T}, s_{0:T}) \| \pi(a_{0:T}, s_{0:T})) = \sum_{t=0}^{T} \mathbb{E}_{s_t, a_t \sim q_\theta}\left[\left(\mathcal{T} \log q_\theta(a_t \mid s_t) - Q^{q_\theta}(s_t, a_t)\right) \nabla_\theta \log q_\theta(a_t \mid s_t)\right].$$

### D.3. Reverse Log-Derivative Trick

Define the unnormalized Boltzmann policy:

$$\pi(a \mid s) \propto \exp(\alpha Q^{q_{\theta^*}}(s, a)).$$

The KL divergence

$$D_{\mathrm{KL}}\big(q_\theta(a_t \mid s_t) \,\|\, \pi(a_t \mid s_t)\big)$$

has, when applying the log-derivative trick, the gradient

$$\nabla_\theta D_{\mathrm{KL}}^{\mathcal{T}} = \mathbb{E}_{a_t \sim q_\theta}\left[(\mathcal{T} \log q_\theta(a_t \mid s_t) - Q^{q_{\theta^*}}(s_t, a_t)) \nabla_\theta \log q_\theta(a_t \mid s_t)\right],$$

which proves the identity.

Importantly, the stop-gradient in $Q^{q_{\theta^*}}$ is not a heuristic modification but arises directly from applying the log-derivative trick in reverse, ensuring that the gradient of the surrogate KL exactly matches the gradient of the original trajectory KL for the current iterate.

Thus, the resulting surrogate loss can be written as:

$$\mathcal{L}_{\mathrm{ME}}(\theta) = \mathcal{T} \sum_{t=0}^{T} \mathbb{E}_{s_t \sim q_{\theta^*}} \left[ D_{\mathrm{KL}} \left( q_\theta(a_t|s_t) \,\Big\|\, \frac{\exp(\alpha Q^{q_{\theta^*}}(s_t, a_t))}{Z(s_t)} \right) \right], \tag{15}$$

and satisfies the following property:

The gradient of the surrogate loss equals the gradient of the original trajectory KL exactly at the current iterate $\theta = \theta^*$.

After the first update, the gradients of the surrogate and true objectives deviate. This mirrors the behavior of most modern RL algorithms—including PPO (Schulman et al., 2017), REPPO (Voelcker et al., 2025) or TRPO (Schulman et al., 2015), which optimize surrogate objectives that are locally exact but globally approximate. This approximation is crucial for computational tractability and is a necessary design choice: otherwise, gradients would need to be computed using samples from $q_\theta$, which is prohibitively expensive because the MDP would have to be unrolled for every gradient update.

### D.4. Connection of the Surrogate Loss to Reinforcement Learning Algorithms

D.4.1. PROXIMAL POLICY OPTIMIZATION (PPO)

Proximal Policy Optimization (PPO) can be derived by applying the log-derivative trick to the Maximum Entropy surrogate objective (see Eq.4) combined with importance sampling. Importance sampling is required because PPO does not learn a state–action value function $Q_\theta$ directly; thus, returns or rewards $R_{\mathrm{env}}(s_t, a_t)$ can only be evaluated at state–action pairs observed under the behavior policy. The resulting surrogate objective can be written as

$$\mathcal{L}_{\mathrm{ME}}(\theta, s_t) = \mathbb{E}_{a_t \sim q_{\theta_{\mathrm{old}}}(a_t|s_t)} \left[ \frac{q_\theta(a_t|s_t)}{q_{\theta_{\mathrm{old}}}(a_t|s_t)} \Big( \mathcal{T} \log q_\theta(a_t|s_t) - Q^{q_{\theta_{\mathrm{old}}}}(s_t, a_t) \Big) \right] + C$$

where $q_{\theta_{\mathrm{old}}}$ denotes the behavior policy used to collect trajectories. As the gradient is computed with respect to $\theta$ this equation effectively implements the log-derivative trick as

$$\begin{aligned} \nabla_\theta \frac{q_\theta(a_t|s_t)}{q_{\theta_{\mathrm{old}}}(a_t|s_t)} &= \nabla_\theta \exp \big( \log q_\theta(a_t|s_t) - \log q_{\theta_{\mathrm{old}}}(a_t|s_t) \big) \\ &= \exp \big( \log q_\theta(a_t|s_t) - \log q_{\theta_{\mathrm{old}}}(a_t|s_t) \big) \nabla_\theta \log q_\theta(a_t|s_t) \\ &= \frac{q_\theta(a_t|s_t)}{q_{\theta_{\mathrm{old}}}(a_t|s_t)} \nabla_\theta \log q_\theta(a_t|s_t) \end{aligned} \tag{16}$$

The PPO objective, however, additionally uses a clipping strategy, where the objective is then given by:

$$\mathcal{L}_{\mathrm{PPO}}(\theta, s_t) = -\mathbb{E}_{a_t \sim q_{\theta_{\mathrm{old}}}(a_t|s_t)} \Big[ \min \Big( r_t(\theta) \, \hat{A}_t, \, \mathrm{clip}(r_t(\theta), 1 - \epsilon, 1 + \epsilon) \, \hat{A}_t \Big) \Big],$$

where $r_t(\theta) = \frac{q_\theta(a_t|s_t)}{q_{\theta_{\mathrm{old}}}(a_t|s_t)}$ are importance weights, $\hat{A}_t$ is an estimate of the advantage function, and $\epsilon$ is the clipping parameter. The advantage function $A_t$ quantifies how much better taking action $a_t$ in state $s_t$ is compared to the expected value of the policy from that state and is defined as $A_t = Q^{\theta_{\mathrm{old}}}(s_t, a_t) - V^{\theta_{\mathrm{old}}}(s_t) - \mathcal{T} \log q_\theta(a_t|s_t)$ where the subtraction of $V^{\theta_{\mathrm{old}}}(s_t)$ reduces the variance of the gradient estimates while leaving the average gradient unchanged.

Compared to the original surrogate loss derived from reverse-KL minimization, PPO samples states $s_t$ from $q_{\theta_{\mathrm{old}}}(s_t)$ rather than from the current variational distribution $q_\theta(s_t)$. Importantly, for the first gradient step, the gradient of the PPO objective coincides exactly with the gradient of the original surrogate loss. For subsequent steps, the gradient begins to deviate, but the clipping mechanism in PPO ensures that this deviation remains controlled and does not become excessively large.

### D.5. Relative Entropy Pathwise Policy Optimization (REPPO)

Relative Entropy Pathwise Policy Optimization (REPPO) can be interpreted as a practical realization of the maximum-entropy reverse-KL objective in Eq. 4, augmented with explicit trust-region control and entropy regularization and can thus

be formalized in the following way:

$$\max_{\theta} \quad \mathcal{L}_{\text{MaxEnt}}(\theta) = \mathcal{T} \sum_{t=0}^{T} \mathbb{E}_{s_t \sim q_{\theta^*}} \left[ D_{\text{KL}}\Big( q_\theta(a_t|s_t) \,\Big\|\, \frac{\exp(\alpha\, Q^{q_{\theta^*}}(s_t, a_t))}{Z(s_t)} \Big) \right],$$

$$\text{subject to} \quad \mathbb{E}_{s_t \sim q_{\theta^*}} \left[ D_{\text{KL}}(q_{\theta^*}(a_t|s_t) || q_\theta(a_t|s_t)) \right] \leq \delta_{\text{KL}} \tag{17}$$

$$\mathbb{E}_{s_t \sim q_{\theta^*}} \left[ \mathcal{H}[q_\theta] \right] \geq \mathcal{H}_{\text{tar}}$$

where $\mathcal{H}[q_\theta]$ is the entropy of $q_\theta$, $\mathcal{H}_{\text{tar}}$ is the target entropy, and the forward KL divergence is used to control for the target KL constraint $\delta_{\text{KL}}$.

**Lagrangian formulation.** The constraints are handled using a Lagrangian relaxation. In REPPO, the entropy Lagrange multiplier is identified with the temperature parameter $\mathcal{T}$, which directly controls the strength of entropy regularization. As a result, only the KL trust-region constraint introduces an additional dual variable $\lambda_{\text{KL}} \geq 0$.

The resulting Lagrangian is given by

$$\mathcal{L}(\theta, \lambda_{\text{KL}}, \mathcal{T}) = \mathcal{T} \sum_{t=0}^{T} \mathbb{E}_{s_t \sim q_{\theta^*}} \left[ D_{\text{KL}}\Big( q_\theta(a_t|s_t) \,\Big\|\, \frac{\exp(\alpha Q^{q_{\theta^*}}(s_t, a_t))}{Z(s_t)} \Big) \right] + \mathcal{T}\mathcal{H}_{\text{tar}}$$

$$+ \lambda_{\text{KL}} \left( \mathbb{E}_{s_t \sim q_{\theta^*}} \left[ D_{\text{KL}}\Big( q_{\theta^*}(a_t|s_t) \,\Big\|\, q_\theta(a_t|s_t) \Big) \right] - \delta_{\text{KL}} \right), \tag{18}$$

where the temperature $\mathcal{T}$ simultaneously acts as the Lagrange multiplier associated with the entropy constraint $\mathbb{E}_{s_t}[\mathcal{H}[q_\theta]] \geq \mathcal{H}_{\text{tar}}$.

**Policy update.** The policy parameters are optimized via gradient ascent on the Lagrangian:

$$\theta_{i+1} = \theta_i + \eta_\theta \nabla_\theta \mathcal{L}(\theta_i, \lambda_{\text{KL}}^i, \mathcal{T}^i), \tag{19}$$

which balances attraction toward the exponentiated $Q$-distribution, a forward-KL trust-region penalty, and entropy regularization controlled by $\mathcal{T}$.

**Non-negativity of Lagrange multipliers.** All Lagrange multipliers in REPPO are required to remain non-negative to ensure valid constraint enforcement. In practice, this is achieved by parameterizing each multiplier via an exponential activation. Concretely, for a free scalar parameter $\omega \in \mathbb{R}$, the corresponding multiplier is given by

$$\lambda = \exp(\omega),$$

which guarantees $\lambda \geq 0$ by construction. Optimization is then performed directly in the unconstrained parameter space of $\omega$, avoiding the need for explicit projection steps while preserving the correctness of the primal–dual formulation.

**Dual updates.** The KL multiplier is updated by gradient ascent:

$$\lambda_{\text{KL}}^{i+1} = \lambda_{\text{KL}}^i - \eta_{\text{KL}} \left( \mathbb{E}_{s_t \sim q_{\theta^*}} \left[ D_{\text{KL}}(q_{\theta^*} \,\|\, q_\theta) \right] - \delta_{\text{KL}} \right) \tag{20}$$

The temperature $\mathcal{T}$ is updated to enforce the target entropy constraint:

$$\mathcal{T}^{i+1} = \mathcal{T}^i - \eta_{\mathcal{T}} \left( \mathbb{E}_{s_t \sim q_{\theta^*}} \left[ \mathcal{H}[q_\theta] \right] - \mathcal{H}_{\text{tar}} \right), \tag{21}$$

where $\eta_{\mathcal{T}}$ denotes the temperature learning rate. Both Lagrange multipliers are parametrized via an exponential mapping, i.e., $\lambda_{\text{KL}} = \exp(\alpha_{\text{KL}})$ and $\mathcal{T} = \exp(\alpha_{\mathcal{T}})$, which guarantees their positivity during optimization.

This formulation yields an adaptive primal–dual optimization scheme in which the trust-region size and entropy level are automatically regulated through $\lambda_{\text{KL}}$ and $\mathcal{T}$, respectively.

**Q-target estimation via TD($\lambda$).** REPPO directly learns an action-value function $Q_\phi(s, a)$ and does not employ a separate state-value function. Consequently, action-value targets are computed using multi-step temporal-difference updates with eligibility traces, i.e. TD($\lambda$).

**Pathwise policy gradients.** REPPO leverages a learned Q-function and the *reparameterization trick* to compute low-variance, pathwise gradients. Actions are generated as

$$a_t = \mu_\theta(s_t) + \epsilon\, \sigma_\theta(s_t) := f_\theta(\xi_t; s_t), \quad \xi_t \sim \mathcal{N}(0, I), \tag{22}$$

allowing gradients to flow directly through sampled actions. The resulting policy-gradient estimator takes the form

$$\nabla_\theta \mathcal{L}_{\text{REPPO}}(\theta, s_t) = \mathbb{E}_{\xi_t}\Big[ \mathcal{T}\, \nabla_\theta \log q_\theta(f_\theta(\xi_t; s_t) \mid s_t) - \nabla_\theta Q^{q_{\theta^*}}(f_\theta(\xi_t; s_t), s_t) \Big]. \tag{23}$$

**KL-aware clipped surrogate objective.** To enforce the KL trust region in practice, REPPO employs a *sample-wise clipped loss* that selects between the reverse-KL surrogate objective and a pure KL penalty, such that the KL penalty is only applied when the KL exceeds a certain threshold. For each sample in a minibatch, the forward KL divergence $\widehat{D}_{\text{KL}}^{(i)}$ for a specific sample $s_t^{(i)}$ in the batch is estimated using $k$ Monte Carlo samples.

The final REPPO surrogate loss can then be written equivalently as

$$\mathcal{L}_{\text{REPPO}}^{\text{clip}}(\theta) = \mathbb{E}_{s_t^{(i)} \sim q_{\theta^*}} \left[ \begin{cases} \mathcal{L}_{\text{ME}}^{(i)}(\theta, s_t^{(i)}), & \text{if } \widehat{D}_{\text{KL}}^{(i)} \leq \delta_{\text{KL}}, \\ \lambda_{\text{KL}}\, \widehat{D}_{\text{KL}}^{(i)}, & \text{if } \widehat{D}_{\text{KL}}^{(i)} > \delta_{\text{KL}}. \end{cases} \right], \tag{24}$$

where $\delta_{\text{KL}}$ is a predefined trust-region threshold, $\lambda_{\text{KL}}$ is a KL penalty coefficient, and $\mathcal{L}_{\text{sur}}^{(i)}(\theta)$ denotes the reverse-KL surrogate loss evaluated at sample $i$.

This adaptive clipping mechanism ensures that policy updates follow the reverse-KL improvement direction when the trust region is respected, while reverting to a pure KL penalty when deviations become too large. As a result, REPPO maintains stable optimization dynamics while retaining the benefits of pathwise gradients and entropy-regularized learning.

**Auxiliary tasks.** REPPO optionally employs auxiliary objectives to stabilize representation learning in sparse-reward regimes and when using small update batches. In particular, it adopts the latent self-prediction auxiliary task introduced in the REPPO paper, which has been shown to significantly improve critic stability and sample efficiency in low-signal settings (see also Jaderberg et al., 2016). Concretely, the critic is augmented with an auxiliary prediction head that encourages the learned latent representation of state–action pairs to be predictive of future representations under the on-policy data distribution. The auxiliary loss is optimized jointly with the critic objective using on-policy rollouts. We refer to the REPPO paper for the precise formulation and architectural details.

**Cross-entropy regression for critic targets.** REPPO further stabilizes critic learning by replacing mean-squared-error regression with a cross-entropy-based regression loss (HL-Gauss; Farebrother et al., 2024), originally inspired by distributional reinforcement learning methods (Bellemare et al., 2017). All critic targets, including those used for policy optimization, are regressed using this loss. This formulation has been shown to yield improved numerical stability and learning dynamics, particularly in deterministic and sparse-reward environments. We refer the reader to the REPPO paper for implementation details and ablation results.

# E. Policy-gradient style derivation for the Log Variance loss

### E.1. Connection between the Log Variance Loss and Off-Policy Surrogate Objectives

In App. E we prove that the surrogate objective in Eq. 4 can also be derived from the *Log Variance (LV) loss* (Richter & Berner, 2024), also known as the *Trajectory Balance* loss in the context of GFlowNets (Malkin et al., 2022a), defined for an off-policy sampling distribution $\omega$ as

$$D_{LV}^\omega(q_\theta(a_{0:T}, s_{0:T+1}) \| \pi(a_{0:T}, s_{0:T+1})) = \frac{1}{2} \mathbb{E}_{(a_{0:T}, s_{0:T+1}) \sim \omega}\left[ \left( \log \frac{q_\theta(a_{0:T}, s_{0:T+1})}{\pi(a_{0:T}, s_{0:T+1})} - b_\theta^\omega \right)^2 \right], \tag{25}$$

where $b_\theta^\omega = \mathbb{E}_{(a_{0:T}, s_{0:T+1},) \sim \omega}[\log \frac{q_\theta(a_{0:T}, s_{0:T+1})}{\pi(a_{0:T}, s_{0:T+1})}]$. Under the assumption that $\omega$ is absolutely continuous with respect to $\pi$ and $q_\theta$, the LV loss is only zero if and only if $\log q_\theta(a_{0:T}, s_{0:T+1}) = \log \pi(a_{0:T}, s_{0:T+1})$. Under these constraints $\omega$ may be *any* distribution over trajectories, such as from on-policy samples or from the replay buffer. Furthermore, the on-policy LV loss yields exactly the same gradient as the rKL loss when $\pi$ does not contain learnable parameters (Richter et al., 2020; Malkin et al., 2022b; Sanokowski et al., 2025b).

We prove that by setting $\omega(s_t, a_t) = q_{\theta^*}(a_t \mid s_t)\mathcal{B}(s_t)$, i.e. for any time step $t$ states are sampled from any off-policy distribution but the actions $a_t$ are sampled on-policy, the LV gradient takes a policy-gradient-like form (see App. E):

$$\mathcal{T}\nabla_\theta D_{LV}^\omega(q_\theta(a_{0:T}, s_{0:T+1}) \| \pi(a_{0:T}, s_{0:T+1}))$$
$$= \sum_{t=0}^{T} \mathbb{E}_{(a_t, s_t) \sim (q_{\theta^*}, \mathcal{B})}\Big[ \Big(\mathcal{T}\log q_\theta(a_t|s_t) + \Big(Q^{\omega, q_{\theta^*}}(s_t, a_t) - V^{\omega, q_{\theta^*}}(s_t)\Big)\Big)\nabla_\theta \log q_\theta(a_t|s_t)\Big], \tag{26}$$

where $Q^{\omega, q_{\theta^*}}(s_t, a_t) = R_{\text{env}}(s_t, a_t) + \mathbb{E}_{s_{t+1}}\Big[V^{\omega, q_{\theta^*}}(s_{t+1})\Big]$ and $V^{\omega, q_{\theta^*}}(s_t) = \delta_{t<T}\, \mathbb{E}_{a_t \sim \omega(a_t|s_t)}\Big[Q^{\omega, q_{\theta^*}}(s_t, a_t) - \mathcal{T}\log q_{\theta^*}(a_t \mid s_t)\Big]$. Since the actions are sampled from $q_{\theta^*}(a_t \mid s_t)$, the *reverse log-derivative trick* can be applied (see App. D.3) to Eq. 26, which yields exactly the off-policy surrogate objective used in **SAC**.

$$\mathcal{L}_{\text{ME}}^{\text{Off-Policy}}(\theta, s_t) = D_{\text{KL}}^{\mathcal{T}}\Big(q_\theta(a_t \mid s_t) \, \Big\| \, \frac{\exp(\alpha\, Q^{\omega, q_{\theta^*}}(s_t, a_t))}{Z(s_t)}\Big). \tag{27}$$

where importantly $s_t \sim \mathcal{B}$.

### E.2. Derivation

**Notation and assumptions**

Consider finite-horizon trajectories written explicitly as

$$(a_{0:T}, s_{0:T+1}) \equiv (s_0, a_0, s_1, a_1, \ldots, s_T, a_T, s_{T+1}).$$

Let $\rho(s_0)$ be the initial state distribution, $q_\theta(a_t \mid s_t)$ the parametric policy (the object we differentiate), and $p(s_{t+1} \mid s_t, a_t)$ the Markov environment dynamics. The reference trajectory distribution $\pi$ is fixed and independent of $\theta$. Remember that

$$q_\theta(a_{0:T}, s_{0:T+1}) = \rho(s_0)\prod_{t=0}^{T} q_\theta(a_t \mid s_t)\, p(s_{t+1} \mid s_t, a_t),$$

$$\pi(a_{0:T}, s_{0:T+1}) = \rho(s_0)\prod_{t=0}^{T} \pi(a_t \mid s_t)\, p(s_{t+1} \mid s_t, a_t),$$

with the key requirement is only that $\pi$ does not depend on $\theta$. Let $\omega$ denote an arbitrary (possibly off-policy) distribution over full trajectories $(a_{0:T}, s_{0:T+1})$. We assume $\omega$ is fixed (does not depend on $\theta$).

Define the per-trajectory log-ratio

$$\ell_\theta(a_{0:T}, s_{0:T+1}) := \log \frac{q_\theta(a_{0:T}, s_{0:T+1})}{\pi(a_{0:T}, s_{0:T+1})} = \sum_{t=0}^{T} \log \frac{q_\theta(a_t \mid s_t)}{\pi(a_t \mid s_t)}$$

The LV loss is defined under the sampling distribution $\omega$ as:

$$D_{LV}^\omega(q_\theta \| \pi) = \frac{1}{2}\,\mathbb{E}_{(a_{0:T}, s_{0:T+1}) \sim \omega}\big[(\ell_\theta(a_{0:T}, s_{0:T+1}) - b_\theta^\omega)^2\big],$$

with the baseline

$$b_\theta^\omega := \mathbb{E}_{(a_{0:T}, s_{0:T+1}) \sim \omega}\big[\ell_\theta(a_{0:T}, s_{0:T+1})\big].$$

Thus, the gradient is given by:

$$\nabla_\theta D_{LV}^\omega = \tfrac{1}{2}\nabla_\theta \mathbb{E}_\omega\big[(\ell_\theta - b_\theta^\omega)^2\big]$$
$$= \mathbb{E}_\omega\big[(\ell_\theta - b_\theta^\omega)(\nabla_\theta \ell_\theta - \nabla_\theta b_\theta^\omega)\big]$$
$$= \mathbb{E}_\omega\big[(\ell_\theta - b_\theta^\omega)\nabla_\theta \ell_\theta\big] - \mathbb{E}_\omega[\ell_\theta - b_\theta^\omega]\,\mathbb{E}_\omega[\nabla_\theta \ell_\theta].$$

By definition $\mathbb{E}_\omega[\ell_\theta - b_\theta^\omega] = 0$, so the second term vanishes. Thus we have the compact unbiased form

$$\boxed{\nabla_\theta D_{LV}^\omega = \mathbb{E}_{(a_{0:T}, s_{0:T+1})\sim\omega}\big[(\ell_\theta(a_{0:T}, s_{0:T+1}) - b_\theta^\omega)\,\nabla_\theta \ell_\theta(a_{0:T}, s_{0:T+1})\big].}$$

Because $\pi$ is fixed and the dynamics $p$ and $\rho$ are independent of $\theta$,

$$\nabla_\theta \ell_\theta(a_{0:T}, s_{0:T+1}) = \nabla_\theta \log q_\theta(a_{0:T}, s_{0:T+1}) = \sum_{t=0}^{T} \nabla_\theta \log q_\theta(a_t \mid s_t).$$

Substituting into the boxed expression and interchanging sums and expectation yields

$$\nabla_\theta D_{LV}^\omega = \mathbb{E}_{(a_{0:T}, s_{0:T+1})\sim\omega}\left[(\ell_\theta(a_{0:T}, s_{0:T+1}) - b_\theta^\omega)\sum_{t=0}^{T-1} \nabla_\theta \log q_\theta(a_t \mid s_t)\right]$$
$$= \sum_{t=0}^{T-1} \mathbb{E}_{(a_{0:T}, s_{0:T+1})\sim\omega}\big[(\ell_\theta(a_{0:T}, s_{0:T+1}) - b_\theta^\omega)\nabla_\theta \log q_\theta(a_t \mid s_t)\big].$$

Next, the total log-ratio $\ell_\theta$ and the baseline $b_\theta^\omega$ are split into "Past" (indices $0$ to $t-1$) and "Future" (indices $t$ to $T$).

$$\ell_\theta(a_{0:T}, s_{0:T+1}) = \sum_{i=0}^{t-1} \log \frac{q_\theta(a_i \mid s_i)}{\pi(a_i \mid s_i)} + \sum_{i=t}^{T} \log \frac{q_\theta(a_i \mid s_i)}{\pi(a_i \mid s_i)} := \ell_\theta(a_{0:t-1}, s_{0:t-1}) + \ell_\theta(a_{t:T}, s_{t:T+1})$$
$$b_\theta^\omega = b_{0:t-1}^\omega + b_{t:T}^\omega := \mathbb{E}_{(a_{0:t}, s_{0:t})\sim\omega}\big[\ell_\theta(a_{0:t-1}, s_{0:t-1})\big] + \mathbb{E}_{(a_{t:T}, s_{t:T+1})\sim\omega(\cdot\mid s_t)\omega(s_t)}\big[\ell_\theta(a_{t:T}, s_{t:T+1})\big]$$

Substitute this into the gradient expression for a single time step $t$:

$$\mathbb{E}_{(a_{0:T}, s_{0:T+1})\sim\omega}\left[\nabla_\theta \log q_\theta(a_t \mid s_t)\Big((\ell_\theta(a_{0:t-1}, s_{0:t-1}) - b_{0:t-1}^\omega) + (\ell_\theta(a_{t:T}, s_{t:T+1}) - b_{t:T}^\omega)\Big)\right].$$

Integrating $\nabla_\theta \log q_\theta(a_t \mid s_t)\,\ell_\theta(a_{0:t-1}, s_{0:t-1})$ over $(a_{0:t-1}, s_{0:t-1})$ then yields:

$$\mathbb{E}_{(a_{t:T}, s_{t:T+1})\sim\omega(\cdot\mid s_t)\omega(s_t)}\left[\nabla_\theta \log q_\theta(a_t \mid s_t)\Big((b_{0:t-1}^\omega - b_{0:t-1}^\omega) + (\ell_\theta(a_{t:T}, s_{t:T+1}) - b_{t:T}^\omega)\Big)\right] \tag{28}$$
$$= \mathbb{E}_{(a_{t:T}, s_{t:T+1})\sim\omega(\cdot\mid s_t)\omega(s_t)}\left[\nabla_\theta \log q_\theta(a_t \mid s_t)\Big((\ell_\theta(a_{t:T}, s_{t:T+1}) - b_{t:T}^\omega\Big), \tag{29}$$

where we have used that $\log q_\theta(a_t \mid s_t)$ does not depend on $(a_{0:t-1}, s_{0:t-1})$.

With the past terms cancelled, only the future $(a_{t:T}, s_{t:T+1})$ components are left:

$$\nabla_\theta D_{LV}^\omega = \sum_{t=0}^{T} \mathbb{E}_{(a_{t:T}, s_{t:T+1})\sim\omega(\cdot\mid s_t)\omega(s_t)}\left[\nabla_\theta \log q_\theta(a_t \mid s_t)\Big((\ell_\theta(a_{t:T}, s_{t:T+1}) - b_{t:T}^\omega)\Big)\right]. \tag{30}$$

This expression can be rewritten by inserting the following definitions:

$$Q^{\omega, q_{\theta^*}}(s_t, a_t) = \mathcal{T}\Big(\log \pi(a_t \mid s_t) - \mathbb{E}_{(a_{t+1:T}, s_{t+1:T+1})\sim\omega(\cdot\mid s_{t+1})}[\ell_\theta(a_{t+1:T}, s_{t+1:T+1})]\Big) = R_{\text{env}}(s_t, a_t) + \mathbb{E}_{s_{t+1}}\Big[V^{\omega, q_{\theta^*}}(s_{t+1})\Big],$$

where

$$V^{\omega,q_{\theta^*}}(s_t) \;=\; \delta_{t<T}\,\mathbb{E}_{a_t\sim\omega(a_t|s_t)}\Big[Q^{\omega,q_{\theta^*}}(s_t,a_t)\;-\;\mathcal{T}\log q_{\theta^*}(a_t\mid s_t)\Big].$$

and $V_t^{\omega,q_{\theta^*}} := \mathbb{E}_{s_t}\Big[V^{\omega,q_{\theta^*}}(s_t)\Big] = -\mathcal{T}\,b_{t:T}^{\omega}.$

$$\mathcal{T}\,\nabla_\theta D_{LV}^{\omega} = \sum_{t=0}^{T}\mathbb{E}_{(a_t,s_t)\sim\omega}\Big[\Big(\mathcal{T}\log q_\theta(a_t\mid s_t) - \Big(Q^{\omega,q_{\theta^*}}(s_t,a_t) - V_t^{\omega,q_{\theta^*}}\Big)\Big)\nabla_\theta\log q_\theta(a_t\mid s_t)\Big].$$

When $\omega(s_t,a_t) = \mathcal{B}(s_t)\,q_{\theta^*}(a_t\mid s_t)$, i.e. at every step $t$ sampling actions $a_t$ is done on policy, any baseline can be used and we can remove any baseline and can thus remove $V_t^{\omega,q_{\theta^*}}$ from the equation and arrive at the policy-gradient LV loss based like update rule given by:

$$\boxed{\mathcal{T}\,\nabla_\theta D_{LV}^{\omega} = \sum_{t=0}^{T}\mathbb{E}_{(a_t,s_t)\sim(q_{\theta^*},\mathcal{B})}\Big[\Big(\mathcal{T}\log q_\theta(a_t\mid s_t) - Q^{\omega,q_{\theta^*}}(s_t,a_t)\Big)\nabla_\theta\log q_\theta(a_t\mid s_t)\Big],}$$

By applying the reverse log derivative trick in reverse (see App. D.3), we finally arrive at Eq. 27

## F. Derivation of Maximum Entropy Wasserstein Policy Optimization

**Functional derivative.**  Expanding the KL gives (up to a constant independent of $q$)

$$\mathcal{L}_{\mathrm{ME}}(q,s) = \int q(a\mid s)\big(\log q(a\mid s) - \alpha Q^{q_{\theta^*}}(s,a)\big)\,da + \mathrm{const},$$

where we write $a$ and $s$ instead of $a_t$ and $s_t$ to improve readability. Hence the functional derivative w.r.t. the density $q$ is

$$\frac{\delta\mathcal{L}(q,s)}{\delta q}(s,a) \;=\; \log q(a\mid s) - \alpha Q^{q_{\theta^*}}(s,a) \;+\; \mathrm{const}. \tag{31}$$

The additive constant (including $\log Z(s)$ and the $+1$ from $\delta\int q\log q/\delta q$) does not affect spatial gradients in $a$ and can therefore be dropped for the flow.

**Wasserstein gradient flow:**  We seek the steepest descent of $\mathcal{J}_s$ in the 2-Wasserstein metric; the corresponding Wasserstein gradient flow (continuity equation form) is (Benamou & Brenier, 2000)

$$\frac{\partial q_\theta(a\mid s)}{\partial t} \;=\; -\nabla_a\cdot\Big(q_\theta(a\mid s)\,\nabla_a\frac{\delta\mathcal{L}(q,s)}{\delta q}(s,a)\Big) \;=\; -\nabla_a\cdot\Big(q_\theta(a\mid s)\,\nabla_a\big(\log q_\theta(a\mid s) - \alpha Q^{q_{\theta^*}}(s,a)\big)\Big), \tag{32}$$

where $v(a) := \nabla_a\frac{\delta\mathcal{L}(q,s)}{\delta q}(s,a)$ and thus we have

$$v(a) \;:=\; \nabla_a\big(\log q_\theta(a\mid s) - \alpha Q^{q_{\theta^*}}(s,a)\big).$$

### F.1. Projection of Wasserstein Flows onto a Parametric Policy Family

To convert the Wasserstein gradient flow in Eq. (32) into a practical update for a parametric policy $q_\theta(a\mid s)$, we project the induced flow onto the space of densities representable by $\theta$. Concretely, we choose the parameter perturbation $\Delta\theta$ that minimizes the KL divergence between the infinitesimally flowed density and the perturbed parametric density (Neklyudov et al., 2023):

$$\Delta\theta = \arg\min_{\delta\theta} D_{\mathrm{KL}}\Big[q_\theta \,\Big\|\, q_\theta + \frac{\partial q_\theta}{\partial t}dt - \nabla_\theta q_\theta\,\delta\theta\Big].$$

Locally, the KL can be approximated by a quadratic form defined by the Fisher information blocks (Pfau et al., 2025):

$$D_{\mathrm{KL}} \approx \begin{pmatrix} dt \\ -\Delta\theta \end{pmatrix}^T \begin{pmatrix} \mathcal{F}_{tt} & \mathcal{F}_{t\theta}^T \\ \mathcal{F}_{t\theta} & \mathcal{F}_{\theta\theta} \end{pmatrix} \begin{pmatrix} dt \\ -\Delta\theta \end{pmatrix},$$

$$\mathcal{F}_{tt} = \mathbb{E}_{q_\theta}[(\partial_t \log q_\theta)^2], \qquad \mathcal{F}_{t\theta} = \mathbb{E}_{q_\theta}[\partial_t \log q_\theta \, \nabla_\theta \log q_\theta], \qquad \mathcal{F}_{\theta\theta} = \mathbb{E}_{q_\theta}[\nabla_\theta \log q_\theta \, \nabla_\theta \log q_\theta^T].$$

Minimizing this quadratic form gives the optimal parameter update

$$\Delta\theta = \mathcal{F}_{\theta\theta}^{-1} \mathcal{F}_{t\theta},$$

where the mixed block $\mathcal{F}_{t\theta}$ captures the correlation between the flow in action space and the parametric gradients. For the reverse–KL functional, $\mathcal{F}_{t\theta}$ reduces to

$$\mathcal{F}_{t\theta} = \mathbb{E}_{a \sim q_\theta}\big[\nabla_\theta \nabla_a \log q_\theta(a \mid s) \, \nabla_a(\log q_\theta(a \mid s) - \alpha Q^{q_{\theta*}}(s,a))\big],$$

which leads directly to the Maximum Entropy WPO update in Eq. (7).

**Projection to parameter space.**   As explained in (Neklyudov et al., 2023; Pfau et al., 2025) the induced flow can be projected on the parametric family $q_\theta$ by minimizing the local KL between the flowed density and the parametric perturbation. The mixed Fisher block is

$$\mathcal{F}_{t\theta} \;=\; \int \nabla_\theta \log q_\theta(a \mid s) \, \frac{\partial q_\theta(a \mid s)}{\partial t} \; da. \tag{33}$$

Insert the flow from Eq. 32:

$$\mathcal{F}_{t\theta} = -\int \nabla_\theta \log q_\theta(a \mid s) \, \nabla_a \cdot \big(q_\theta(a \mid s)\, v(a)\big) \, da.$$

Expand the divergence and apply integration by parts. Using $\nabla_a \cdot (qv) = q\,\nabla_a \cdot v + v \cdot \nabla_a q$ we obtain

$$\mathcal{F}_{t\theta} = -\int \nabla_\theta \log q_\theta \Big(q_\theta \nabla_a \cdot v + v \cdot \nabla_a q_\theta\Big) da$$

$$= -\int \nabla_\theta q_\theta \, \nabla_a \cdot v \, da \;-\; \int \nabla_\theta \log q_\theta \, v \cdot \nabla_a q_\theta \, da. \tag{34}$$

For the first integral we apply the divergence theorem (integration by parts):

$$-\int \nabla_\theta q_\theta(a \mid s)\, \nabla_a \cdot v(a)\, da = -\int_{\partial\Omega} \big(\nabla_\theta q_\theta(a \mid s)\, v(a)\big) \cdot n(a)\, dS(a) + \int \nabla_a\big(\nabla_\theta q_\theta(a \mid s)\big) \cdot v(a)\, da, \tag{35}$$

where $\Omega = \mathbb{R}^n$ is the action space and the surface integral is the boundary term. Under the standard regularity / tail-decay assumptions for parametric policies (e.g. Gaussian tails, or other densities for which $\nabla_\theta q_\theta$ vanishes sufficiently fast at $|a| \to \infty$) the surface integral vanishes. With the boundary term dropped we continue:

Thus we arrive at:

$$\mathcal{F}_{t\theta} = \int \nabla_a\big(\nabla_\theta q_\theta\big) \cdot v \, da - \int \nabla_\theta \log q_\theta \, v \cdot \nabla_a q_\theta \, da.$$

Using the identities

$$\nabla_a q_\theta = q_\theta \, \nabla_a \log q_\theta, \qquad \nabla_\theta q_\theta = q_\theta \, \nabla_\theta \log q_\theta,$$

we expand the first integrand:

$$\nabla_a(\nabla_\theta q_\theta) = \nabla_a\big(q_\theta \nabla_\theta \log q_\theta\big) = (\nabla_a q_\theta)\, \nabla_\theta \log q_\theta + q_\theta \, \nabla_a \nabla_\theta \log q_\theta.$$

Thus

$$\int \nabla_a(\nabla_\theta q_\theta) \cdot v \, da = \int \Big[(\nabla_a q_\theta)\,(\nabla_\theta \log q_\theta \cdot v) + q_\theta\,(\nabla_a \nabla_\theta \log q_\theta \cdot v)\Big] da.$$

Hence the terms including $(\nabla_a q_\theta)\,(\nabla_\theta \log q_\theta \cdot v)$ cancel and we arrive at:

$$\mathcal{F}_{t\theta} = \int q_\theta(a \mid s)\,(\nabla_a \nabla_\theta \log q_\theta(a \mid s) \cdot v(a))\, da.$$

Divide by $q_\theta$ and write the integral as an expectation:

$$\mathcal{F}_{t\theta} = \mathbb{E}_{a \sim q_\theta(\cdot|s)}[\nabla_\theta \nabla_a \log q_\theta(a \mid s) \cdot v(a)], \tag{36}$$

where importantly $\nabla_\theta$ only acts on $\nabla_a \log q_\theta(a \mid s)$ and not on $v(a)$. Using $v(a) = \nabla_a(\log q_\theta(a \mid s) - \alpha Q^{q_{\theta*}}(s,a))$, we obtain

$$\boxed{\mathcal{F}_{t\theta} = \mathbb{E}_{a \sim q_\theta(\cdot|s)}\big[\nabla_\theta \nabla_a \log q_\theta(a \mid s) \, \nabla_a\big(\log q_\theta(a \mid s) - \alpha Q^{q_{\theta*}}(s,a)\big)\big].}$$

**Final Update Formula:** Let $\mathcal{F}_{\theta\theta}$ denote the Fisher information matrix

$$\mathcal{F}_{\theta\theta} = \mathbb{E}_{a \sim q_\theta(\cdot|s)}[\nabla_\theta \log q_\theta \, \nabla_\theta \log q_\theta^\top].$$

Thus the parameter increment

$$\Delta\theta = \mathcal{T}\mathcal{F}_{\theta\theta}^{-1}\mathcal{F}_{t\theta}.$$

is given by:

$$\boxed{\theta_{t+1} = \theta_t + \eta \, \mathcal{F}_{\theta\theta}^{-1} \, \mathbb{E}_{s \sim q_{\theta*}, \, a \sim q_\theta}\left[\Big(\nabla_\theta \nabla_a \log q_\theta(a \mid s)\Big)\Big(\nabla_a\big(Q^{q_{\theta*}}(s,a) - \mathcal{T}\log q_\theta(a \mid s)\big)\Big)\right].} \tag{37}$$

Where we have included the outer scaling $\mathcal{T}$ and averaging over states sampled from $q_{\theta*}$ (the stop-gradient sampling distribution). The corresponding surrogate loss can then be written as:

$$\boxed{\mathcal{L}_{\text{WPO}}(\theta, s_t) = \frac{\mathcal{T}}{2} \mathbb{E}_{a_t \sim q_{\theta*}}\left[\Big\|\nabla_{a_t}\Big(\log q_\theta(a_t|s_t) - \alpha \, Q^{q_{\theta*}}(s_t, a_t)\Big)\Big\|^2\right],} \tag{38}$$

where the gradient of Eq. 38 yields

$$\nabla_\theta \mathcal{L}_{\text{WPO}}(\theta, s_t) = \mathbb{E}_{a_t \sim q_{\theta*}}\left[\Big(\nabla_\theta \nabla_{a_t} \log q_\theta(a_t \mid s_t)\Big)\Big(\nabla_{a_t}\big(\mathcal{T}\log q_\theta(a_t \mid s_t) - Q^{q_{\theta*}}(s_t, a_t)\big)\Big)\right].$$

**Practical approximations.** Equation (37) requires the mixed derivative $\nabla_\theta \nabla_a \log q_\theta$ and a (possibly large) Fisher matrix inverse $\mathcal{F}_{\theta\theta}^{-1}$. In practice, we (as in (Pfau et al., 2025)) approximate expectations with samples, and use tractable approximations to $\mathcal{F}_{\theta\theta}^{-1}$ (diagonal, block diagonal, K-FAC, or other). The heuristic scaling used in (Pfau et al., 2025) (e.g. scaling of $\nabla_\mu$ and $\nabla_\sigma$ when backpropagating through a Gaussian policy) can be applied when $q_\theta$ is a simple gaussian distribution.

## G. Policy Gradient Theorem for Diffusion Policies

To rewrite the KL objective in Eq. 10 in a form suitable for the direct application of the policy-gradient theorem, we introduce a modified reward that absorbs both the diffusion-consistency (log-ratio) terms and, at $k = 1$, the environment reward:

$$\tilde{R}_{\text{DME}}(a_t^{k-1}, a_t^k, s_t) := \begin{cases} -\mathcal{T}\log\dfrac{q_\theta(a_t^{k-1} \mid a_t^k, s_t)}{\pi(a_t^k \mid a_t^{k-1}, s_t)} & \text{if } k > 1, \\[4mm] R_{\text{env}}(s_t, a_t^0) - \mathcal{T}\log\dfrac{q_\theta(a_t^{k-1} \mid a_t^k, s_t)}{\pi(a_t^k \mid a_t^{k-1}, s_t)} & \text{if } k = 1. \end{cases} \tag{39}$$

The forward-diffusion distribution $\pi(a_t^k \mid a_t^{k-1}, s_t)$ does not depend on $\theta$. Furthermore, for any $q_\theta$ we have the identity

$$\mathbb{E}_{a \sim q_\theta}[\nabla_\theta \log q_\theta(a)] = 0.$$

Therefore the term $\log q_\theta(a_t^{k-1} \mid a_t^k, s_t)$ that appears inside the modified reward may be replaced by

$$\log q_{\theta*}(a_t^{k-1} \mid a_t^k, s_t),$$

where $\theta^*$ indicates a *stop–gradient* (i.e., treated as a constant). Under this convention the reward $\tilde{R}_{\text{DME}}$ is independent of the policy parameters $\theta$.

Consequently, the KL objective can be written as an expectation of a parameter-independent per-step reward. Thus $\tilde{R}_{\text{DME}}$ is a valid reinforcement-learning reward signal, and the classical policy-gradient theorem applies directly without further modification or justification.

### G.1. Resulting policy gradient

Applying the policy-gradient theorem to the parameter-independent reward $\tilde{R}_{\text{DME}}$, we obtain

$$\nabla_\theta D_{\text{KL}}^{\mathcal{T}} = - \sum_{t=0, k=K}^{T,1} \mathbb{E}_{a_t^{k-1}, s_t} \left[ \left( \tilde{R}_{\text{DME}}(a_t^{k-1}, \tilde{s}_{\tilde{t}}) + \mathbb{E}_{\tilde{s}_{\tilde{t}+1}} \left[ V_{\text{DME}}^{q_\theta}(\tilde{s}_{\tilde{t}+1}) \right] \right) \nabla_\theta \log q_\theta(a_t^{k-1} \mid a_t^k, s_t) \right]. \tag{40}$$

where

$$Q_{\text{DME}}^{q_{\theta^*}}(\tilde{s}_{\tilde{t}}, a_t^{k-1}) = \tilde{R}_{\text{DME}}(a_t^{k-1}, \tilde{s}_{\tilde{t}}) + \mathbb{E}_{\tilde{s}_{\tilde{t}+1}} \left[ V_{\text{DME}}^{q_{\theta^*}}(\tilde{s}_{\tilde{t}+1}) \right],$$

$$V_{\text{DME}}^{q_{\theta^*}}(\tilde{s}_{\tilde{t}}) = \delta_{t<T} \, \mathbb{E}_{a_t^{k-1} \sim q_{\theta^*}(\cdot | \tilde{s}_{\tilde{t}})} \left[ Q_{\text{DME}}^{q_{\theta^*}}(\tilde{s}_{\tilde{t}}, a_t^{k-1}) - \mathcal{T} \log \frac{q_{\theta^*}(a_t^{k-1}|a_t^k, s_t)}{\pi_{\theta^*}(a_t^k|a_t^{k-1}, s_t)} \right], \tag{41}$$

and $\tilde{s}_{\tilde{t}} = \tilde{s}_{\tilde{t}(t,k)} = (s_t, a_t^k, k)$.

By inserting the above definitions, we obtain:

$$\nabla_\theta D_{\text{KL}}^{\mathcal{T}} = \sum_{t,k} \mathbb{E}_{a_t^{k-1}, s_t} \left[ \left( \mathcal{T} \log \frac{q_\theta(a_t^{k-1} \mid a_t^k, s_t)}{\pi(a_t^k \mid a_t^{k-1}, s_t)} - Q_{\text{DME}}^{q_{\theta^*}}(\tilde{s}_{\tilde{t}}, a_t^{k-1}) \right) \nabla_\theta \log q_\theta(a_t^{k-1} \mid a_t^k, s_t) \right].$$

### G.2. Reverse Log-Derivative Trick and Diffusion Surrogate Loss

We now derive the surrogate loss used in Eq. 11.

Consider the KL divergence

$$\mathcal{L}_{\text{DME}}(\theta, \tilde{s}_{\tilde{t}}) = \mathcal{T} D_{\text{KL}} \left( q_\theta(a_t^{k-1} \mid \tilde{s}_{\tilde{t}}) \, \big\| \, \pi(a_t^k \mid a_t^{k-1}, s_t) \frac{\exp \left( \alpha \, Q_{\text{Diff}}^{q_\theta}(\tilde{s}_{\tilde{t}}), a_t^{k-1} \right)}{Z(\tilde{s}_{\tilde{t}})} \right) \right)$$

$$= \mathbb{E}_{a_t^{k-1}} \left[ \mathcal{T} \log \frac{q_\theta(a_t^{k-1} \mid \tilde{s}_{\tilde{t}})}{\pi(a_t^k \mid a_t^{k-1}, s_t)} - Q_{\text{DME}}^{q_\theta}(\tilde{s}_{\tilde{t}}, a_t^{k-1}) \right] + Z(\tilde{s}_{\tilde{t}}),$$

where $a_t^{k-1} \sim q_\theta(a_t^{k-1} \mid \tilde{s}_{\tilde{t}})$ and $Z(\tilde{s}_{\tilde{t}})$ is the normalizing partition function (independent of $\theta$).

Differentiating w.r.t. $\theta$ and applying the log-derivative trick gives

$$\nabla_\theta \mathcal{L}_{\text{DME}}(\theta, \tilde{s}_{\tilde{t}}) = \mathbb{E}_{a_t^{k-1}} \left[ \left( \mathcal{T} \log \frac{q_\theta(a_t^{k-1} \mid \tilde{s}_{\tilde{t}})}{\pi(a_t^k \mid a_t^{k-1}, s_t)} - Q_{\text{DME}}^{q_\theta}(\tilde{s}_{\tilde{t}}, a_t^{k-1}) \right) \nabla_\theta \log q_\theta(a_t^{k-1} \mid \tilde{s}_{\tilde{t}}) \right]. \tag{42}$$

This identity, applied at each diffusion index $k$ and each timestep $t$, yields the surrogate loss expression in Eq. 11 of the main text:

$$\mathcal{L}_{\text{DME}}(\theta) = \mathcal{T} \sum_{t,k} \mathbb{E}_{s_t \sim q_{\theta^*}} \left[ D_{\text{KL}} \left( q_\theta(\cdot \mid \tilde{s}_{\tilde{t}}) \, \big\| \, \pi(a_t^k \mid \cdot, s_t) \frac{\exp(\alpha Q_{\text{DME}}^{q_{\theta^*}}(\tilde{s}_{\tilde{t}}, \cdot))}{Z(\tilde{s}_{\tilde{t}})} \right) \right]$$

This completes the derivation.

### G.3. Generalization to Learnable Diffusion Coefficients

The derivation presented above assumes that only the reverse diffusion kernels $q_\theta(a_t^{k-1} \mid a_t^k, s_t)$ are parameterized and learned. However, as discussed in Sec. 2.4, our model additionally learns the diffusion coefficients $\beta$ at each diffusion step. As a consequence, the forward diffusion kernel $\pi_\theta(a_t^k \mid a_t^{k-1}, s_t)$ also depends on learnable parameters and must be optimized jointly with the reverse process.

While this setting slightly departs from the assumptions made in the earlier derivation, the extension is straightforward. In particular, it can be shown that the following surrogate objective remains valid:

$$D_{\mathrm{KL}}^{\mathcal{T}} = \mathcal{T} \sum_{t,k} \mathbb{E}_{s_t \sim q_{\theta^*}} \left[ D_{\mathrm{KL}} \left( q_\theta(\cdot \mid a_t^k, s_t) \,\middle\|\, \pi_\theta(a_t^k \mid \cdot, s_t) \frac{\exp(\alpha Q_{\mathrm{DME}}^{q_{\theta^*}}(s_t, \cdot))}{Z(s_t, a_t^k)} \right) \right].$$

To see this, we start from the reverse-KL objective

$$D_{\mathrm{KL}}^{\mathcal{T}}(q_\theta \| \pi_\theta) = \mathbb{E}_{s_t \sim q_\theta} \left[ \sum_{t,k} \left( \mathcal{T} \log \frac{q_\theta(a_t^{k-1} \mid a_t^k, s_t)}{\pi(a_t^k \mid a_t^{k-1}, s_t)} - R_{\mathrm{env}}(s_t, a_t^0) \mathbf{1}_{\{k=1\}} \right) \right].$$

Compared to the fixed-forward case, the key difference is that $\pi_\theta$ now depends on the policy parameters. Consequently, when differentiating the objective, gradients must be taken not only with respect to $q_\theta$ but also through the forward diffusion model $\pi_\theta$. Applying the policy-gradient theorem yields

$$\nabla_\theta D_{\mathrm{KL}}^{\mathcal{T}} = \sum_{t,k} \mathbb{E}_{s_t, a_t^k, a_t^{k-1} \sim q_\theta} \left[ \left( \mathcal{T} \log \frac{q_\theta(a_t^{k-1} \mid a_t^k, s_t)}{\pi(a_t^k \mid a_t^{k-1}, s_t)} - R_{\mathrm{env}}(s_t, a_t^0) \mathbf{1}_{\{k=1\}} \right) \nabla_\theta \log q_\theta(a_t^{k-1} \mid a_t^k, s_t) \right. \tag{43}$$

$$\left. - \mathcal{T} \nabla_\theta \log \pi_\theta(a_t^k \mid a_t^{k-1}, s_t) \right]. \tag{44}$$

Crucially, this gradient can equivalently be obtained by optimizing the surrogate loss

$$\mathcal{L}_{\mathrm{DME}}(\theta) = \mathcal{T} \sum_{t,k} \mathbb{E}_{s_t, a_t^k \sim q_{\theta^*}} \left[ D_{\mathrm{KL}} \left( q_\theta(\cdot \mid a_t^k, s_t) \,\middle\|\, \pi_\theta(a_t^k \mid \cdot, s_t) \frac{\exp(\alpha Q_{\mathrm{DME}}^{q_{\theta^*}}(s_t, \cdot))}{Z(s_t, a_t^k)} \right) \right],$$

by applying the log-derivative trick with respect to $q_\theta$. This shows that the proposed surrogate objective remains valid even when the forward diffusion process is learned, and that jointly optimizing the reverse transitions and diffusion coefficients yields the correct policy gradient.

### G.4. Diffusion Maximum Entropy PPO

To incorporate PPO-style trust-region updates, also to learned forward diffusion kernels, the masking (introduced by the clipping and the min operation) has to be applied to both the reverse and forward diffusion kernel.

We can write this masking operation as:

$$\mathbf{m}_{t,k}(\theta) = \begin{cases} 1, & \rho_{t,k}(\theta) \hat{A}_{t,k} < \tilde{\rho}_{t,k}(\theta) \hat{A}_{t,k} \ \lor \ (\rho_{t,k}(\theta) = \tilde{\rho}_{t,k}(\theta)), \\ 0, & \text{otherwise}, \end{cases} \tag{45}$$

where

$$\rho_{t,k}(\theta) = \frac{q_\theta(a_t^{k-1} \mid a_t^k, s_t)}{q_{\theta_{\mathrm{old}}}(a_t^{k-1} \mid a_t^k, s_t)} \tag{46}$$

denote the importance ratio at diffusion step $k$ (with $\theta_{\mathrm{old}}$ the behavior policy used to collect data), $\tilde{\rho}_{t,k}(\theta)$ is the clipped ratio and let $\hat{A}_{t,k}$ denote the corresponding unnormalized advantage signal.

The final gradient can then be written as

$$\nabla_\theta D_{\mathrm{KL}}^{\mathcal{T}} = \sum_{t,k} \mathbb{E}_{s_t, a_t^k, a_t^{k-1} \sim q_{\theta_{\mathrm{old}}}} \left[ \rho_{t,k}(\theta)\, \mathbf{m}_{t,k}(\theta) \left( \hat{A}_{t,k}\ \nabla_\theta \log q_\theta(a_t^{k-1} \mid a_t^k, s_t) - \mathcal{T}\, \nabla_\theta \log \pi_\theta(a_t^k \mid a_t^{k-1}, s_t) \right) \right].$$

### G.5. Diffusion Maximum Entropy Wasserstein Policy Optimization

In the following we we derive how to compute the Wasserstein Gradient Flow in parameter space, when also diffusion coefficients $\beta$ are learned. Since we have already derived the Wasserstein Gradient flow for a variation in $q$ in App. F, we will now derive the Wasserstein Gradient flow for a variation in $\pi$ (see also (Neklyudov et al., 2023)):

$$F(\pi) \;=\; D_{\mathrm{KL}}(q\|\pi) \;=\; \int_{\mathbb{R}^d} q(x) \log \frac{q(x)}{\pi(x)}\, dx \;=\; \underbrace{\int q \log q\, dx}_{\text{const in } \pi} - \int q(x) \log \pi(x)\, dx.$$

Let $\pi_\varepsilon = \pi + \varepsilon \eta$ with $\int \eta\, dx = 0$ (mass-preserving variations). Then

$$\frac{d}{d\varepsilon} F(\pi_\varepsilon)\Big|_{\varepsilon=0} = -\int q(x) \frac{\eta(x)}{\pi(x)}\, dx = \int \left( -\frac{q(x)}{\pi(x)} \right) \eta(x)\, dx.$$

Hence the first variation (functional derivative) is

$$\frac{\delta F}{\delta \pi}(x) \;=\; -\frac{q(x)}{\pi(x)}.$$

(Any additive constant is irrelevant for the $W_2$ flow since it disappears under $\nabla$.)

The 2-Wasserstein gradient flow of F is the continuity equation

$$\partial_t \pi_t \;=\; \nabla \cdot \left( \pi_t\, \nabla \frac{\delta F}{\delta \pi} \right) \;=\; \nabla \cdot \left( \pi_t\, \nabla \left( -\frac{q}{\pi_t} \right) \right).$$

Equivalently, in velocity-field form with

$$v_t \;=\; -\nabla \left( \frac{\delta F}{\delta \pi} \right) \;=\; \nabla \left( \frac{q}{\pi_t} \right),$$

the flow is

$$\partial_t \pi_t \;+\; \nabla \cdot (\pi_t v_t) \;=\; 0. \tag{47}$$

Analogously to App. F we have to compute:

$$\mathcal{F}_{t\theta} \;=\; \int \nabla_\theta \log \pi_\theta(a)\, \frac{\partial \pi_\theta(a)}{\partial t}\, da.$$

By using the identity from Eq. 36, and by inserting Eq. 47 we arrive at:

$$\mathcal{F}_{t\theta} = \mathbb{E}_{a \sim \pi_\theta(\cdot)}[\nabla_\theta \nabla_a \log \pi_\theta(a) \cdot v(a)]$$

$$= \nabla_\theta \mathbb{E}_{a \sim \pi_{\theta^*}(\cdot)}\left[\nabla_a \log \pi_\theta(a) \cdot \nabla_a \frac{q(a)}{\pi_{\theta^*}(a)}\right]$$

$$= \nabla_\theta \int \pi_{\theta^*}(a) \nabla_a \log \pi_\theta(a) \cdot \nabla_a \frac{q(a)}{\pi_{\theta^*}(a)} \; da$$

$$= -\nabla_\theta \int q(a) \nabla_a \log \pi_\theta(a) \cdot \nabla_a \log \frac{\pi_{\theta^*}(a)}{q(a)} \; da$$

$$= -\nabla_\theta \int q(a) \nabla_a \log \pi_\theta(a) \cdot \left(\nabla_a \log \pi_{\theta^*}(a) - \nabla_a \log q(a)\right) \; da$$

$$= \nabla_\theta \int q(a) \nabla_a \log \pi_\theta(a) \cdot \left(\nabla_a \log q(a) - \nabla_a \log \pi_\theta(a)\right) \; da$$

$$= \nabla_\theta \frac{1}{2} \int q(a) \left\| \nabla_a \log q(a) - \nabla_a \log \pi_\theta(a) \right\|^2 \; da$$

$$= \nabla_\theta \frac{1}{2} \mathbb{E}_{a \sim q(a)}\left[\left\|\nabla_a(\log q(a) - \log \pi_\theta(a))\right\|^2\right].$$

where we have used that $\nabla_a \frac{q(a)}{\pi_{\theta^*}(a)} = -\frac{q(a)}{\pi_{\theta^*}(a)} \nabla_a \log \frac{\pi_{\theta^*}(a)}{q(a)}$.

**Combining the Wasserstein Gradient Flow for the Forward and Reverse Diffusion Kernels:**

The surrogate loss for the Wasserstein Gradient Flow, when applied to both the forward and reverse diffusion kernels, follows an identical mathematical structure. Specifically, the loss for the forward kernel is given by

$$\mathcal{L}_{\pi_\theta} = \mathbb{E}_{a \sim q(a)}\left[\left\|\nabla_a(\log q(a) - \log \pi_\theta(a))\right\|^2\right],$$

while the loss for the reverse kernel is expressed as

$$\mathcal{L}_{q_\theta} = \mathbb{E}_{a \sim q(a)}\left[\left\|\nabla_a(\log q_\theta(a) - \log \pi(a))\right\|^2\right].$$

Combining their gradients and combining them into a surrogate loss yields:

$$\mathcal{L}_{\pi_\theta, q_\theta} = \mathbb{E}_{a \sim q(a)}\left[\left\|\nabla_a(\log q_\theta(a) - \log \pi_\theta(a))\right\|^2\right].$$

To adapt this to the diffusion process, we substitute $q(a)$ with $q_\theta(a_t^{k-1}|a_t^k, s_t)$ and $\pi_\theta(a)$ with $\pi_\theta(a_t^k|a_t^{k-1}, s_t)\frac{\exp(\alpha Q_{\text{DME}}^{q_{\theta^*}}(\tilde{s}_{\tilde{t}}, a_t^{k-1}))}{Z(\tilde{s}_{\tilde{t}})}$. This substitution yields the final form of the loss function:

$$\mathcal{L}_{\text{DME-WPO}}(\theta, \tilde{s}_{\tilde{t}}) = \frac{\mathcal{T}}{2} \mathbb{E}_{a_t^{k-1}}\left[\left\|\nabla_{a_t^{k-1}}\left(\log \frac{q_\theta(a_t^{k-1} \mid a_t^k, s_t)}{\pi_\theta(a_t^k \mid a_t^{k-1}, s_t)} - \alpha Q_{\text{DME}}^{q_{\theta^*}}(\tilde{s}_{\tilde{t}}, a_t^{k-1})\right)\right\|^2\right]. \tag{48}$$

where $a_t^{k-1} \sim q_{\theta^*}(a_t^{k-1}|a_t^k, s_t)$ and thus its gradient reads as:

$$\nabla_\theta \mathcal{L}_{\text{DME-WPO}}(\theta, \tilde{s}_{\tilde{t}}) = \mathbb{E}_{a_t^{k-1}}\left[\left(\nabla_\theta \nabla_{a_t^{k-1}} \log \frac{q_\theta(a_t^{k-1}|a_t^k, s_t)}{\pi_\theta(a_t^k|a_t^{k-1}, s_t)}\right)\left(\nabla_{a_t^{k-1}}\left(\mathcal{T}\log \frac{q_\theta(a_t^{k-1}|a_t^k, s_t)}{\pi_\theta(a_t^k|a_t^{k-1}, s_t)} - Q_{\text{DME}}^{q_{\theta^*}}(\tilde{s}_{\tilde{t}}, a_t^{k-1})\right)\right)\right],$$
$$\tag{49}$$

This derivation is strictly only correct if $q_\theta(a_t^{k-1}|a_t^k, s_t)$ and $\pi_\theta(a_t^k|a_t^{k-1}, s_t)$ do not have shared parameters, which is true by the design of our diffusion samplers, where $\pi_\theta(a_t^k|a_t^{k-1}, s_t)$ only depends on $\beta_{k-1}$ and not on $\theta'$ and $q_\theta(a_t^{k-1}|a_t^k, s_t)$ depends on $\theta'$ and $\beta_k$ (recall that $\theta = (\theta', \{\beta_K, \ldots, \beta_0\})$, see Sec. 2.4). When the forward and reverse diffusion kernels have shared parameters, the derivation might need to be adapted by assuming that a variation of $q$ has an effect on $\pi$.

# H. Hyperparameters and Implementation Details

## H.1. Diffusion-based Hyperparameters:

For DME-WPO, DME-PPO, and DME-REPPO, we use a learnable linear diffusion schedule and a non-learnable prior distribution with standard deviation of 3.0. In REPPO-DIME, we follow (Celik et al., 2025) and use a learnable cosine diffusion schedule and a non-learnable prior distribution with standard deviation of 2.2.

## H.2. Adjusting Reinforcement Learning Hyperparameters

In reinforcement learning (RL), the discount factor $\gamma$ and the trace decay parameter $\lambda$ play a central role in determining the effective planning horizon and credit assignment. In diffusion-augmented Markov Decision Processes (MDPs), the environment dynamics are expanded over $K$ diffusion steps, which effectively increases the horizon by a factor of $K$. To preserve comparable temporal discounting behavior, it is therefore necessary to appropriately rescale these hyperparameters.

We propose adjusting the discount factor according to

$$\gamma_{\text{DME}} = \gamma^{\frac{1}{K}} \, ,$$

and applying the same transformation to $\lambda$, yielding $\lambda_{\text{DME}} = \lambda^{\frac{1}{K}}$. In all experiments, we use $\gamma = 0.999$ and $\lambda = 0.98$ with $K = 8$ diffusion steps.

**Binning Bounds for Standard MDPs.** Adapting $\gamma_{\text{DME}}$ also affects the value range hyperparameters $v_{\min}$ and $v_{\max}$ used in the HL-gauss loss (Farebrother et al., 2024), which define the binning bounds. We first review how these bounds are derived in the standard MDP setting.

Consider an infinite-horizon discounted MDP with discount factor $\gamma \in [0, 1)$ and bounded rewards

$$r_t \in [r_{\min}, r_{\max}] \, .$$

The return from time step $t$ is defined as

$$G_t = \sum_{k=0}^{\infty} \gamma^k r_{t+k} \, .$$

Since rewards are bounded, the return is bounded by the corresponding geometric series. The maximal return is achieved when $r_{t+k} = r_{\max}$ for all $k$, giving

$$G_t \leq \sum_{k=0}^{\infty} \gamma^k r_{\max} = \frac{r_{\max}}{1 - \gamma} \, .$$

Similarly, the minimal return is obtained when $r_{t+k} = r_{\min}$ for all $k$:

$$G_t \geq \sum_{k=0}^{\infty} \gamma^k r_{\min} = \frac{r_{\min}}{1 - \gamma} \, .$$

The action–value function of a policy $\pi$ is defined as

$$Q^\pi(s, a) = \mathbb{E}_\pi[G_t \mid s_t = s, \, a_t = a] \, .$$

Since expectations preserve bounds, it follows that

$$\boxed{Q^\pi(s, a) \in \left[ \frac{r_{\min}}{1 - \gamma}, \, \frac{r_{\max}}{1 - \gamma} \right] \, .}$$

In maximum-entropy reinforcement learning, the reward additionally includes a policy log-probability term and is therefore formally unbounded. Nevertheless, these bounds remain effective in practice provided that the temperature parameter is sufficiently small.

**Binning Bounds for Diffusion-Based MDPs.** We now consider the sparse-reward structure induced by diffusion-based MDPs (see Sec. 3), in which rewards are zero for $K-1$ consecutive time steps and non-zero rewards occur only every $K$-th step. Formally, rewards satisfy

$$r_{t+j} = 0 \quad \text{for } j \not\equiv K - 1 \, (\text{mod } K),$$

and when a reward occurs,

$$r_{t+j} \in [r_{\min}, r_{\max}].$$

Under this structure, the return from time step $t$ can be written as

$$G_t = \sum_{n=0}^{\infty} \gamma^{nK+(K-1)} r_{t+nK+(K-1)} \, .$$

The maximal return is achieved when every available reward equals $r_{\max}$:

$$G_t \leq \gamma^{K-1} r_{\max} \sum_{n=0}^{\infty} \gamma^{nK} = \frac{\gamma^{K-1} r_{\max}}{1 - \gamma^K} \, .$$

Similarly, the minimal return is bounded by

$$G_t \geq \frac{\gamma^{K-1} r_{\min}}{1 - \gamma^K} \, .$$

Consequently, the action–value function satisfies

$$Q^\pi(s, a) \in \left[ \frac{\gamma^{K-1} r_{\min}}{1 - \gamma^K}, \ \frac{\gamma^{K-1} r_{\max}}{1 - \gamma^K} \right] \, .$$

Finally, substituting $\gamma \to \gamma_{\text{DME}} = \gamma^{\frac{1}{K}}$ yields

$$Q^\pi(s, a) \in \left[ \frac{\gamma^{-1} r_{\min}}{1 - \gamma}, \ \frac{\gamma^{-1} r_{\max}}{1 - \gamma} \right] \, .$$

Since $\gamma$ is typically chosen very close to 1, this rescaling only slightly expands the binning range, indicating that the standard bounds remain largely appropriate for diffusion-based MDPs.

### H.3. Temperature Tuning in DME-REPPO and DME-WPO

For automatic temperature tuning as in (Haarnoja et al., 2018), we follow (Celik et al., 2025) and use a lower bound of the entropy to control the temperature.

The entropy lower bound $H_{\text{lower}} \leq H(\pi(a_t|s_t))$ is given

$$H_{\text{lower}} = \mathbb{E}_{a_t^{0:K} \sim q_\theta} \left[ \sum_{k=K}^{1} \log \frac{\pi(a_t^k|a_t^{k-1}, s_t)}{q_\theta(a_t^{k-1}|a_t^k, s_t)} - \log q(a_t^{(K)}, s_t) \right]. \tag{50}$$

In practice, the summation over reverse diffusion steps

$$\sum_{k=K}^{1} \log \frac{\pi(a_t^k \mid a_t^{k-1}, s_t)}{q_\theta(a_t^{k-1} \mid a_t^k, s_t)} \tag{51}$$

is not computed explicitly. Instead, we estimate this quantity using Monte Carlo sampling over diffusion time steps. Concretely, we sample a diffusion index $k \sim \mathcal{U}(\{1, \dots, K\})$ and construct the unbiased estimator

$$\sum_{k=K}^{1} \log \frac{\pi(a_t^k \mid a_t^{k-1}, s_t)}{q_\theta(a_t^{k-1} \mid a_t^k, s_t)} \approx K \, \mathbb{E}_{k \sim \mathcal{U}(\{1, \dots, K\})} \left[ \log \frac{\pi(a_t^k \mid a_t^{k-1}, s_t)}{q_\theta(a_t^{k-1} \mid a_t^k, s_t)} \right], \tag{52}$$

where $(a_t^k, a_t^{k-1})$ are obtained from the forward diffusion process at the sampled timestep $k$. This Monte Carlo estimator avoids the need to simulate the full reverse diffusion chain for every update, significantly reducing computational overhead while preserving an unbiased estimate of the original summation.

Following Celik et al. (2025), we regulate only the diffusion-dependent log-ratio term

$$\sum_{k=K}^{1} \log \frac{\pi(a_t^k \mid a_t^{k-1}, s_t)}{q_\theta(a_t^{k-1} \mid a_t^k, s_t)}, \tag{53}$$

as the prior term $-\log q(a_t^{(K)}, s_t)$ is constant and therefore does not affect the temperature optimization. In our experiments, this diffusion log-ratio term is tuned to a target entropy value of 2.2. We employ a higher target entropy than in DIME, as our policy does not apply a $\tanh$ transformation to the action outputs, resulting in a higher effective action entropy.

### H.4. Auxiliary Loss for Diffusion-Based MDPs

#### H.4.1. AUXILIARY SELF-PREDICTION LOSSES

For DME-REPPO and DME-WPO, we follow REPPO and employ auxiliary self-prediction losses (Jaderberg et al., 2016) on the reward and state embeddings to improve learning in environments with sparse rewards.

In this section, we first explain auxiliary losses as used in REPPO and then explain how we extended them to diffusion MDPs.

**Reward prediction.** The auxiliary reward prediction loss trains the model to predict the immediate reward from the embedding of the current state–action pair. Given a learned reward predictor $f_r(\cdot)$, the loss is defined as

$$\mathcal{L}_{\text{rew}} = \mathbb{E}_{(s_t, a_t, r_t) \sim \mathcal{D}} \left[ \left\| f_r(\phi(s_t, a_t)) - r_t \right\|_2^2 \right], \tag{54}$$

where $\phi(s_t, a_t)$ is the embedding of the state-action pair, and $\mathcal{D}$ denotes the replay buffer. This objective encourages the learned representations to encode features that are predictive of task rewards, thereby facilitating more efficient policy optimization.

**Embedding prediction.** To further capture environment dynamics, a self-prediction loss for the embedding of the next state is employed. Specifically, an embedding transition predictor $f_e(\cdot)$ is trained to predict the embedding of the next state $\phi(s_{t+1})$ given the embedding of the current state and action:

$$\mathcal{L}_{\text{emb}} = \mathbb{E}_{(s_t, a_t, s_{t+1}, a_{t+1}) \sim \mathcal{D}} \left[ \left\| f_e(\phi(s_t, a_t)) - \phi(s_{t+1}, a_{t+1}) \right\|_2^2 \right]. \tag{55}$$

This auxiliary objective encourages the model to learn representations that reflect the underlying transition dynamics.

**Overall objective.** The auxiliary losses are combined with the main RL objective using weighting coefficients $\kappa$:

$$\mathcal{L}_{\text{total}} = \mathcal{L}_{\text{RL}} + \kappa \left( \mathcal{L}_{\text{rew}} + \mathcal{L}_{\text{emb}} \right). \tag{56}$$

These auxiliary tasks are used only during training and have no effect on policy execution at inference time. In REPPO, we set $\kappa = 1$. For WPO and ME-WPO, this value was reduced to $0.1$. This adjustment is necessary because WPO and ME-WPO employ an $L_2$ regression loss on gradients, which operates on a different scale than the KL-based loss.

#### H.4.2. AUXILIARY SELF-PREDICTION LOSSES FOR DIFFUSION MDPS

In diffusion-based MDPs, actions are generated through a sequence of reverse diffusion steps, yielding intermediate diffusion actions $\{a_t^K, ..., a_t^0\}$. This structure naturally induces additional temporal and diffusion-level dependencies that can be exploited through auxiliary self-prediction objectives. We extend the auxiliary losses introduced in Section H.4.1 with diffusion-aware prediction tasks that encourage consistency across diffusion steps and environment transitions.

**Diffusion embedding transition prediction.** We introduce an auxiliary loss that predicts the embedding of the next sampled state–diffusion action pair $(\phi(\tilde{s}_{\tilde{t}+1}, a_t^{k-1}))$ given the embedding of the previous pair $(\phi(\tilde{s}_{\tilde{t}}, a_t^k))$ at a randomly sampled diffusion step $k$. Let $f_d(\cdot)$ denote a diffusion embedding transition predictor. The corresponding loss is defined as

$$\mathcal{L}_{\text{diff}} = \mathbb{E}_{(\tilde{s}_{\tilde{t}}, a_t^k, a_{t+1}^k) \sim \mathcal{D}, \, k \sim \mathcal{U}(\{1, \ldots, K\})} \left[ \left\| f_d(\phi(\tilde{s}_{\tilde{t}}, a_t^k)) - \phi(\tilde{s}_{\tilde{t}}, a_t^{k-1}) \right\|_2^2 \right]. \tag{57}$$

This objective encourages the learned representations to capture the diffusion action embeddings that evolve across environment transitions.

**Final diffusion embedding step prediction.** At the final diffusion step $k = 1$, the diffusion action corresponds to the executed environment action. We therefore introduce an additional auxiliary loss that predicts the embedding of the next state–action pair $(\phi(s_{t+1}, a_{t+1}^0))$ from the embedding of the previous pair $(\phi(s_t, a_t^0))$:

$$\mathcal{L}_{\text{final}} = \mathbb{E}_{(\tilde{s}_{\tilde{t}}, a_t^0, \tilde{s}_{\tilde{t}+1}, a_{t+1}^0) \sim \mathcal{D}} \left[ \left\| f_f(\phi(\tilde{s}_{\tilde{t}}, a_t^0)) - \phi(\tilde{s}_{\tilde{t}+1}, a_{t+1}^0) \right\|_2^2 \right]. \tag{58}$$

This loss directly regularizes the representations used by the executed policy, ensuring temporal consistency at the level of environment action embeddings.

**Overall auxiliary objective.** The diffusion-specific auxiliary losses are combined with the standard auxiliary objectives and the main RL loss:

$$\mathcal{L}_{\text{total}} = \mathcal{L}_{\text{RL}} + \kappa \left( \eta \, \mathcal{L}_{\text{diff}} + (1 - \eta)(\mathcal{L}_{\text{rew}} + \mathcal{L}_{\text{final}}) \right). \tag{59}$$

where $\kappa \geq 0$ and $0 \leq \eta \leq 1$.

In our experiments with **DME-WPO**, we found that setting $\kappa = 0.02$ and $\eta = 0.98$ yields decent performance. For **DME-REPPO**, the working values were $\kappa = 0.25$ and $\eta = 0.98$. For both methods, we observed that setting $\eta = 1.0$, which effectively disables $\mathcal{L}_{\text{diff}}$, led to inconsistent results across different random seeds. Instead, choosing a value for $\eta$ slightly below 1.0 resulted in more reliable performance. Surprisingly, within the range of hyperparameters we tested, this modified diffusion MDP auxiliary loss proved to be more robust and reliable for **DME-WPO** than for **DME-REPPO**.

