# OpenReview forum: "A Framework for Diffusion-Based Maximum Entropy Reinforcement Learning Algorithms"
_ICML.cc/2026/Conference — Submitted to ICML 2026_

### Official Review · Reviewer_E8RG · 2026-03-08

**Soundness:** 4
**Presentation:** 4
**Significance:** 3
**Originality:** 3
**Overall Recommendation:** 5
**Confidence:** 3

**Summary:**

The paper frames Maximum Entory RL as diffusion-based sampling from the optimal Boltzmann policy. The core contirbution is minimizing reverse KL between a diffusion policy and this optimal policy, applying the data processing inequality twice to obtain a tractable upper bound. This bound decomposes over both environment time steps and diffusion denoising steps, yielding an augmented MDP where denosing steps are intermediate states. This framework plugs into common algorightms like PPO, REPPO with only minor modifications.

**Compliance With Llm Reviewing Policy:**

Affirmed.

**Key Questions For Authors:**

1. Minor suggestion on visualization. The line colors in Figure 1 are difficult to distinguish. Also, currently line styles are split into two groups. Grouping into three categories, proposed algorithms, their corresponding baselines, and other baselines would make comparisons much easier to parse.
2. This is out of curiosity. ME-RL explicitly requires stochastic policies, and SDE naturally provides this. The KL decomposition in Eq. 10 relies on forward and reverse processes having well-defined stochastic transition kernels, making the KL between Gaussian transitions tractable, a structure ODE flows lack. However, ODE offers significantly faster inference. Would an ODE-based variant be feasible, perhaps by injecting stochasticity only at the final step or through a hybrid formulation?

**Limitations:**

yes

**Strengths And Weaknesses:**

## Strengths
1. Clean theoretical development. Reverse KL is well motivated for RL, it avoids wasting mass on suboptimal actions and only require samples from diffusion policies. While reverse KL can suffer from mode collapse, diffusion policies mitigate this tendency through iterative denoising that can route probability toward multiple modes. Although applying the data processing inequality twice could yield a loose bound, the derivation is clear and well-structured.

2. Loss decomposes per diffusion step. Since the KL splits into a sum over individual transitions, one can sample random (t,k) paris and backprop through a single denoising step, avoiding the need to store the full k-step computation grach in mempry.

3. Generality across algoirhtms. The same augmented MDP and diffusion-aware Q-function plug into fundamentally different policy optimization strategies with minimal changes, demonstrated through DME-PPO (log-derivative trick + clipping), DME-REPPO (reparameterization + trust region), and DME-WPO (Wasserstein score matching).

## Weakness
1. Runtime overhead. Experimental gains over non-diffusion baselines are moderate on standard benchmarks, while computational costs are significant.
2. Limited benchmarks. Evaluations cover only DeepMind control tasks, with no high-dimensional or explicitly multimodal tasks where diffusion's expressiveness would be most justified. Also no ablation on the $k$,  which directly affects runtime, bound tightness and Q-function learning difficulty.

---

> ### Author Rebuttal · Authors · 2026-03-28
>
> We thank the reviewer for the positive assessment of the theoretical development, the reverse-KL motivation, and the generality of the framework. We appreciate the constructive comments on evaluation and presentation. Below we address the main points.
>
> **A1: W2 - No experiments on multimodal tasks.**
> To address this concern, we added a toy multimodal control example that directly tests whether the compared methods can learn multimodal action distributions. A visualization of the learned behaviors is provided here ([gif](https://github.com/dmerlicml/DMERL_Rebuttal/blob/main/MultimodalActions/multi_agents.gif)), and the corresponding per-state action histograms are shown here ([histograms](https://github.com/dmerlicml/DMERL_Rebuttal/blob/main/MultimodalActions/histo.png)).
>
> In this environment, a 2D agent outputs actions in $[-1,1]$, corresponding to relative movement direction changes between $-90^\circ$ and $90^\circ$. The reward is a double-well potential with two optima at $-45^\circ$ and $45^\circ$, making the optimal policy inherently multimodal. The state is the agent’s current orientation. After an action is taken, the transition maps it to the nearest global maximum of the double-well reward potential, while the reward is still computed from the original, unrounded action. This yields only 8 possible states, allowing us to visualize for each state the learned action histogram together with the reward landscape.
>
> We compare **REPPO, DIME-REPPO, DPPO, DME-REPPO, DME-WPO, and DME-PPO** on this task. **REPPO** suffers from mode collapse because its Gaussian policy can represent only a single mode; consequently, the agent consistently moves toward the lower-left corner regardless of initialization. **DPPO** [2], which corresponds to DME-PPO at temperature zero, also fails to represent the multimodal action distribution, leading the agent to move in cycles. By contrast, **DIME-REPPO** and our methods, namely **DME-REPPO, DME-WPO, and DME-PPO**, all assign probability mass to both reward maxima for each state, demonstrating successful learning of multimodal action distributions and substantially richer behavior.
>
> While this experiment is intentionally simple, it provides direct evidence that our approach can model and exploit multimodal action structure, unlike unimodal baselines and DPPO, which is another diffusion-based baseline.
>
> **A2: Q1 - Figure 1 visualization.**
> We thank the reviewer for this suggestion and have incorporated the feedback in the updated figure: [link](https://github.com/dmerlicml/DMERL_Rebuttal/blob/main/IQM/all_envs_methods_iqm_eval_return.png).
>
> **A3: Q2 - ODE-based variants / faster inference.**
> This is an interesting question, and we agree that ODE-based variants are a natural direction to consider.
>
> A maximum-entropy RL formulation for ODE-based samplers appears possible, since the process is not fully deterministic: the initial condition is still sampled from a prior distribution. The corresponding log-density can be written as
> $
> \log p_\theta(x) = \log p_0(x_0) - \int_0^1 \nabla \cdot u_\theta(x_t) dt,
> \quad
> x_t = x_0 + \int_0^t u_s(x_s) ds.
> $
> Thus, the log-probability of a sample decomposes over flow-integration steps, conceptually similar to how standard Gaussian policies decompose over time or the joint probability distribution of our diffusion policies decomposes over diffusion steps. Based on this, we believe that the policy-gradient theorem could directly be applied on the left-hand side of Eq. 1 in our paper, which would result in a surrogate loss defined over individual flow-integration steps. However, computing $\nabla \cdot u_\theta(x_t)$ is expensive, so one would likely need Hutchinson trace estimators. This is our current mathematical intuition, but the details would need to be worked out more carefully.
>
> ODE-based formulations may also be possible through approaches such as in [3], where flow samplers are trained with physics-informed losses [1]. At the same time, such flow-based samplers have empirically been found to perform weakly even on simple low-dimensional tasks [4].
>
> Finally, we would also like to note that, after training, it is already possible to sample from the diffusion model by simulating the corresponding probability-flow ODE [5].
>
> We thank the reviewer again for the helpful comments. We believe these revisions improve the updated manuscript of this paper.
>
> **References**
> [1] Raissi, M., et al. *Physics-informed neural networks: A deep learning framework for solving forward and inverse problems involving nonlinear partial differential equations.* JCP 2019.
>
> [2] Ren, A. Z., et al. *Diffusion Policy Policy Optimization.* ICLR 2025.
>
> [3] Tian, Y., et al. *Liouville flow importance sampler.* ICML 2024.
>
> [4] He, J., et al. *No Trick, No Treat: Pursuits and Challenges Towards Simulation-free Training of Neural Samplers.* ICML Workshop 2025.
>
> [5] Song, Y., et al. *Maximum likelihood training of score-based diffusion models.* NeurIPS 2021.

---

> > ### Author Rebuttal · Reviewer_E8RG · 2026-04-06
> >
> > Thank you for the rebuttal and the discussion on the ODE. I will keep my current score.

---

### Official Review · Reviewer_o2Q3 · 2026-03-09

**Soundness:** 1
**Presentation:** 3
**Significance:** 3
**Originality:** 2
**Overall Recommendation:** 3
**Confidence:** 4

**Summary:**

This paper proposes a unified framework for diffusion-based maximum-entropy reinforcement learning (DME-RL). The main idea is to view RL as a sampling problem, then parameterize the policy with a reverse diffusion process so it can represent complex, multimodal action distributions more flexibly than standard policies.

**Compliance With Llm Reviewing Policy:**

Affirmed.

**Key Questions For Authors:**

See the weaknesses above.

**Limitations:**

Yes.

**Strengths And Weaknesses:**

Strengths:
The paper provides a fairly unified framework for incorporating expressive diffusion policies into maximum-entropy RL, with diffusion-based variants of PPO/REPPO/WPO that seem practically implementable with relatively small changes. It is also well motivated and empirically promising, since diffusion policies can model complex multimodal action distributions and the reported benchmark results are strong.

---

Weaknesses:

It seems to me that the theory parts need more rigor.

- **The policy-gradient derivation is presented too strongly for the actual parameter-dependent objective.**
  Because the diffusion reverse-KL objective contains parameter-dependent log-ratio terms, this is not a literal application of the classical policy-gradient theorem for parameter-independent rewards. Appendix G obtains the result by a gradient-preserving rewrite that replaces the $\log q_\theta$ contribution with a stop-gradient version $q_{\theta^\star}$, thereby defining a surrogate objective to which the standard theorem can be applied. This supports a local first-order identity, but it should be described as a tailored surrogate derivation rather than as the classical theorem applying directly to the original objective. In the learnable-forward setting, where $\pi_\theta$ also depends on $\theta$, this distinction is even more important.

- **The treatment of learnable forward diffusion kernels is not cleanly integrated into the theorem statement.**
  In Sec. 2.4 and Sec. 3, the paper explicitly makes the diffusion coefficients learnable and folds them into $\theta$, so that the forward kernel is written as $\pi_\theta$ and depends on the learnable schedule. However, around Eq. (39) in Appendix G, the derivation states that the forward diffusion distribution does not depend on $\theta$. That assumption is only compatible with the fixed-forward-kernel case. To the paper's credit, Appendix G.3 later acknowledges this and derives an amended gradient with an additional
  $$
  -\,T \nabla_\theta \log \pi_\theta(a_t^k \mid a_t^{k-1}, s_t)
  $$
  term. But this means the exposition is internally inconsistent unless the reader notices that the earlier derivation is only a special case and that the learnable-forward-kernel correction is deferred to G.3. This restriction or amendment should be made explicit already in the main theorem statement.

- **Proposition 2.1 is more conditional than the prose around it suggests, and the connection to SAC is overstated.**
  Proposition 2.1 starts from the log-variance objective in Eq. (5) and derives the surrogate in Eq. (6), but the statement already assumes a hybrid sampling construction in which states are drawn from an off-policy distribution $B$ while actions are sampled on-policy from $q_{\theta^\ast}(\cdot \mid s)$. This is a fairly special setup, not a generic equivalence between the LV objective and the usual off-policy actor loss. Moreover, the paper itself notes a few lines later that the LV objective is *not* an $f$-divergence and does *not* satisfy the same information-theoretic properties as the reverse KL. For that reason, the conclusion's claim of a ``direct equivalence'' to the SAC-style loss is stronger than what the body really establishes. At minimum, the proposition should state its assumptions more explicitly, and the conclusion should be weakened to a conditional correspondence rather than a direct equivalence.

- **The DME-WPO derivation is not fully convincing once forward and reverse kernels share learnable parameters.**
  Equation (14) is motivated by projecting the Wasserstein flow of Eq. (11) using reverse-KL terms for the reverse kernel and forward-KL terms for the forward kernel. But earlier the paper states that the forward and reverse processes share learnable parameters through the coefficients $\beta_k$. Once these kernels are coupled through shared parameters, one would expect cross-dependencies to appear in the gradient. These are not discussed in the main derivation around Eq. (14). As a result, it is unclear whether Eq. (14) is the exact projected Wasserstein gradient for the coupled model, or rather a practical surrogate motivated by the uncoupled derivation.

- **The connection to WPO becomes weaker once the natural-gradient component is removed.**
  In Sec. 2.3, the ME-WPO derivation emphasizes that projected Wasserstein flows should be preconditioned by the inverse Fisher matrix. But in the diffusion setting, the paper then states around Eq. (14) that the natural-gradient update from WPO cannot be applied and that explicit preconditioning is omitted in all experiments. This weakens the claim that DME-WPO is a faithful diffusion generalization of WPO rather than a heuristic surrogate inspired by it. If the preconditioning step is essential to the original WPO interpretation, then the paper should explain more clearly what theoretical guarantee, if any, remains after dropping it.

- The discussion immediately after Eq. (1) claims that minimizing the right-hand side of (1) *optimizes* the variational policy and that reducing the upper bound *tightens the gap* between the two divergences. This is not rigorous as, in general, obtaining a smaller upper bound does *not* by itself show that the bound has become tighter.

---

> ### Author Rebuttal · Authors · 2026-03-28
>
> We thank the reviewer for the careful reading and thoughtful technical comments. Below we clarify the main theoretical points and how we will revise the paper.
>
> **A1: W1 - Policy-gradient derivation.** \
> In maximum-entropy RL, it is well known that a term of the form $\log q_\theta$ may appear in the reward and the policy gradient theorem still applies (see [1], Sec. 4.1). As we show in App. D.2, L.832ff, $E_{a_t \sim q_\theta} [\nabla_\theta \log q_\theta(a_t | s_t)]=0.$ Hence we may equivalently write $E_{a_t \sim q_\theta}[\nabla_\theta \log q_\theta(a_t |s_t)]=0 = E_{a_t \sim q_\theta}[\nabla_\theta \log q_{\theta^\star}(a_t | s_t)],$ with $\theta^\star$ denoting stop-gradient parameters. Therefore, $\tilde{r}(s_t,a_t):=r(s_t,a_t)- \mathcal{T} \log q_{\theta^*}(a_t\mid s_t)$ can be treated as parameter-independent in the policy-gradient derivation, since the gradient of the $\log q_\theta$ term vanishes in expectation.
>
> **A2: W2 - Learnable forward kernels.**
> We thank the reviewer for pointing this out. As noted, in App. G we first derive the policy gradient theorem for diffusion MDPs with fixed forward kernels, and then later generalize it to learnable forward kernels in App. G.3. We agree this should be stated explicitly at the beginning of App. G, and we will do so in the revision.
>
> **A3: W3 - connection to SAC and the LV objective.**
> We agree that “direct equivalence” in L.436ff is too strong and will revise it to “correspondence.” Elsewhere in the paper we already use the weaker framing of a connection/correspondence (see L.059ff and Sec. 2.2), and we thank the reviewer for highlighting this inconsistency.
>
> We also want to stress why this result is interesting. Since the LV objective is not an $f$-divergence, its off-policy use does not inherit the usual information-theoretic guarantees of reverse-KL objectives. We believe this is interesting in light of the errata B in [2], where off-policy actor-critic convergence is not established for the general learned-policy setting, but only for the tabular case. Our correspondence between the LV loss and off-policy RL may help explain this: because LV is not an $f$-divergence, the upper-bound argument no longer applies, and minimizing LV does not necessarily minimize the KL between the policy marginal and the optimal distribution in Eq. (1). We will add this discussion in the revised manuscript.
>
> **A4: W4 - DME-WPO with learnable forward/reverse kernels.**
> We understand the concern. As discuss it in App. G.5, L.1535ff this coupling issue is avoided by construction: the diffusion coefficients are parameterized with distinct parameters for each diffusion step, so the problematic shared-parameter cross-dependencies do not arise in the form described by the reviewer. Thus in Eq. 48, $\log q_\theta$ depends on the diffusion coefficient $\beta_k$ at step $k$, while $\log \pi_\theta$ depends on the coefficient $\beta_{k-1}$ at step $k-1$. We agree this assumption should be made more explicit and will do so in the revision. If one instead used a shared parameterization over diffusion time, additional cross-terms would appear and the derivation would need to be modified.
>
> **A5: W5 - Relation to WPO after dropping explicit natural-gradient preconditioning.**
> This is a very good point, which we briefly discuss in L.293ff. We do not claim that our implementation is an exact realization of the original WPO natural-gradient update. In WPO [3], the inverse Fisher is itself only approximated as they apply natural Fisher updates with respect to the mean and standard deviation of the Gaussian policy (see L. 189 ff right). As discussed in App. F.1, L.1296ff the exact inverse Fisher with respect to network parameters is intractable and thus practical approximations must be used, e.g. KFAC or Gaussian approximations. We argue in L.293ff. that since the inverse Fisher arises from a Taylor expansion of the KL divergence, an additional KL regularization loss as we use in DME-WPO can be interpreted as an alternative preconditioning mechanism.
>
> **A6: W6 - Discussion after Eq. (1): upper bound vs. tighter bound.**
> We agree that the current wording is not accurate enough. Minimizing an upper bound does not by itself imply that the bound has become tighter. The correct statement is: reducing the upper bound either shrinks the bound itself or reduces the objective being bounded. We meant to say: If the target objective does not decrease while the bound decreases, then the gap must shrink. We will revise the text accordingly.
>
> Overall, we thank the reviewer for his comments, as they help us sharpen the theoretical presentation in the revised manuscript.
>
> [1] Levine, Sergey. *Reinforcement learning and control as probabilistic inference: Tutorial and review.* arXiv:1805.00909.
>
> [2] Degris, Thomas, Martha White, and Richard S. Sutton. *Off-policy actor-critic.* arXiv:1205.4839.
>
> [3] Pfau, David, et al. "Wasserstein policy optimization." arXiv preprint arXiv:2505.00663.

---

> > ### Author Rebuttal · Reviewer_o2Q3 · 2026-04-03
> >
> > Thank you for the rebuttal. I will keep my current score as the theoretical issues seem hard to be fully resolved during the rebuttal (though they may be fixable after a major revision). For example, for the policy-gradient issue (W1), the response points to a stop-gradient / surrogate-style rewrite. That may justify a first-order-equivalent surrogate under certain conditions, but it still does not show that the manuscript’s stronger statement — namely, that the classical policy-gradient theorem applies directly to the original parameter-dependent objective — is correct. So even after reading the rebuttal, it remains unclear to me how the current presentation is mathematically justified.

---

> > > ### Author Response · Authors · 2026-04-03
> > >
> > > We agree the presentation can be improved, but the gradient identity is correct and is standard in maximum-entropy RL; see [1], Sec. 4.1 or [2], App. A.1.
> > >
> > > Consider $$
> > > D^\tau_{\rm KL}(q_\theta(a_{0:T},s_{0:T+1})\|\pi(a_{0:T},s_{0:T+1}))
> > > \stackrel{C}{=}
> > > \sum_{t=0}^T
> > > \mathbb E_{(s_t,a_t)\sim q_\theta}
> > > [\tau\log q_\theta(a_t|s_t)-R_{\rm env}(s_t,a_t)].$$
> > > Let
> > > $\tilde r_\theta(s_t,a_t):=\tau\log q_\theta(a_t|s_t)-R_{\rm env}(s_t,a_t).$
> > > Then
> > > $
> > > D^\tau_{\rm KL}\stackrel{C}{=}\sum_{t=0}^T\mathbb E_{(s_t,a_t)\sim q_\theta}[\tilde r_\theta(s_t,a_t)].
> > > $
> > > Differentiating,
> > > $$
> > > \nabla_\theta D^\tau_{\rm KL}
> > > \stackrel{C}{=}
> > > \sum_{t=0}^T
> > > \mathbb E_{(s_t,a_t)\sim q_\theta}
> > > [\tilde r_\theta(s_t,a_t)\nabla_\theta\log q_\theta(s_t,a_t)]
> > > +
> > > \sum_{t=0}^T
> > > \mathbb E_{(s_t,a_t)\sim q_\theta}
> > > [\nabla_\theta \tilde r_\theta(s_t,a_t)].
> > > $$
> > > Since $R_{\rm env}$ is independent of $\theta$,
> > > $$
> > > \nabla_\theta \tilde r_\theta(s_t,a_t)=\tau\nabla_\theta\log q_\theta(a_t|s_t).
> > > $$
> > > Moreover,
> > > $$\mathbb E_{a_t\sim q_\theta(\cdot|s_t)}[\nabla_\theta\log q_\theta(a_t|s_t)]=\int q_\theta(a_t|s_t)\nabla_\theta\log q_\theta(a_t|s_t)\,da_t=\int \nabla_\theta q_\theta(a_t|s_t)da_t=\nabla_\theta 1=0.$$
> > > Hence
> > > $\mathbb E_{(s_t,a_t)\sim q_\theta}[\nabla_\theta \tilde r_\theta(s_t,a_t)]=0, $ and therefore
> > > $$\nabla_\theta D^\tau_{\rm KL}\stackrel{C}{=}\sum_{t=0}^T\mathbb E_{(s_t,a_t)\sim q_\theta}[\tilde r_\theta(s_t,a_t)\nabla_\theta\log q_\theta(s_t,a_t)].$$
> > > Thus $\tilde r_\theta$ contributes no extra gradient term, since its derivative vanishes in expectation.
> > >
> > > As we argue in App. G.3, with learnable forward kernel parameters, the derivation is the same except that the forward kernel contributes an additional explicit gradient term, which is also present in the surrogate loss. Starting from
> > > $$
> > > D^\tau_{\rm KL}(q_\theta\|\pi_\theta)\stackrel{C}{=}\sum_{t,k}\mathbb E_{q_\theta}\Big[\tau\log\frac{q_\theta(a_t^{k-1}|a_t^k,s_t)}{\pi_\theta(a_t^k|a_t^{k-1},s_t)}-R_{\rm env}(s_t,a_t^0)\delta_{k1}\Big],$$
> > > we obtain
> > > $$\nabla_\theta D^\tau_{\rm KL}=\sum_{t,k}\mathbb E_{q_\theta}\Big[
> > > g_\theta\nabla_\theta\log q_\theta(s_t,a_t^k,a_t^{k-1})
> > > \Big]+\sum_{t,k}\mathbb E_{q_\theta}[\nabla_\theta g_\theta],$$
> > > where
> > > $$
> > > g_\theta=
> > > \tau\log\frac{q_\theta(a_t^{k-1}|a_t^k,s_t)}
> > > {\pi_\theta(a_t^k|a_t^{k-1},s_t)}-R_{\rm env}(s_t,a_t^0)\delta_{k1}.$$
> > > Now
> > > $$
> > > \nabla_\theta g_\theta=\tau\nabla_\theta\log q_\theta(a_t^{k-1}|a_t^k,s_t)-\tau\nabla_\theta\log\pi_\theta(a_t^k|a_t^{k-1},s_t).$$
> > > Using again
> > > $$
> > > \mathbb E_{q_\theta}[\nabla_\theta\log q_\theta(a_t^{k-1}|a_t^k,s_t)]=0,
> > > $$
> > > this becomes
> > > $$
> > > \nabla_\theta D^\tau_{\rm KL}=\sum_{t,k}\mathbb E_{q_\theta}\Big[g_\theta\nabla_\theta\log q_\theta(s_t,a_t^k,a_t^{k-1})\Big]-\tau\sum_{t,k}\mathbb E_{q_\theta}[\nabla_\theta\log\pi_\theta(a_t^k|a_t^{k-1},s_t)].
> > > $$
> > > So compared to the fixed-forward case, the only new term is
> > > $-\tau\nabla_\theta\log\pi_\theta(a_t^k|a_t^{k-1},s_t).$
> > >
> > > The same surrogate remains valid because it contains the explicit term $-\log\pi_\theta$. Here and below, $q:=q_{\theta^*}$:
> > > $$L_{\rm DME}(\theta) = \tau \sum_{t,k} \mathbb E_{s_t,a_t^k\sim q} \Big[D_{\rm KL}\Big(q_\theta(\cdot|a_t^k,s_t)\|\pi_\theta(a_t^k|\cdot,s_t)\frac{\exp(\alpha Q^{q}_{\rm DME}(s_t,\cdot))}{Z}\Big)\Big].$$
> > >
> > > Expanding the KL,
> > > $$L_{\rm DME}(\theta)\stackrel{C}{=}\tau\sum_{t,k}\mathbb E_{s_t,a_t^k\sim q}\mathbb E_{a_t^{k-1}\sim q_\theta(\cdot|a_t^k,s_t)}\Big[\log q_\theta(a_t^{k-1}|a_t^k,s_t)-\log\pi_\theta(a_t^k|a_t^{k-1},s_t)-\alpha Q^{q}_{\rm DME}(s_t,a_t^{k-1})\Big].$$
> > >
> > > Differentiating,
> > > $$\nabla_\theta L_{\rm DME}(\theta)\stackrel{C}{=}\tau\sum_{t,k}\mathbb E_{s_t,a_t^k\sim q}\Big[\mathbb E_{a_t^{k-1}\sim q_\theta}[h_\theta\nabla_\theta\log q_\theta(a_t^{k-1}|a_t^k,s_t)]+\mathbb E_{a_t^{k-1}\sim q_\theta}[\nabla_\theta h_\theta]\Big],$$
> > >
> > > with
> > >
> > > $h_\theta=\log q_\theta(a_t^{k-1}|a_t^k,s_t)-\log\pi_\theta(a_t^k|a_t^{k-1},s_t)-\alpha Q^{q}_{\rm DME}(s_t,a_t^{k-1}).$
> > >
> > > Since $Q^{q}_{\rm DME}$ is defined with a stop_gradient on $q$,
> > >
> > > $$\nabla_\theta h_\theta=\nabla_\theta\log q_\theta(a_t^{k-1}|a_t^k,s_t)-\nabla_\theta\log\pi_\theta(a_t^k|a_t^{k-1},s_t).$$
> > >
> > > Using again
> > >
> > > $$\mathbb E_{a_t^{k-1}\sim q_\theta}[\nabla_\theta\log q_\theta(a_t^{k-1}|a_t^k,s_t)]=0,$$
> > >
> > > we get
> > >
> > > $$\nabla_\theta L_{\rm DME}(\theta)\stackrel{C}{=}
> > > \tau\sum_{t,k}\mathbb E_{s_t,a_t^k\sim q}
> > > \mathbb E_{a_t^{k-1}\sim q_\theta}
> > > [h_\theta\nabla_\theta\log q_\theta(a_t^{k-1}|a_t^k,s_t)]
> > > -\tau\sum_{t,k}\mathbb E_{s_t,a_t^k\sim q}
> > > \mathbb E_{a_t^{k-1}\sim q_\theta}
> > > [\nabla_\theta\log\pi_\theta(a_t^k|a_t^{k-1},s_t)].$$
> > >
> > > This is exactly the same structure as above: the usual policy-gradient term through $q_\theta$, plus the additional correction
> > > $-\tau\nabla_\theta\log\pi_\theta(a_t^k|a_t^{k-1},s_t).$
> > > Hence the surrogate gradient matches the gradient of the learned-forward objective exactly.
> > >
> > > [1] Levine, Sergey. Reinforcement learning and control as probabilistic inference: Tutorial and review.
> > >
> > > [2] Abdolmaleki, Abbas, et al. "Maximum a Posteriori Policy Optimisation." ICLR, 2018.

---

### Official Review · Reviewer_MgQ2 · 2026-03-09

**Soundness:** 3
**Presentation:** 3
**Significance:** 2
**Originality:** 2
**Overall Recommendation:** 3
**Confidence:** 3

**Summary:**

The paper proposes a general reinforcement learning framework (DME-RL) in which the policy is parameterized by a diffusion model and trained under the maximum entropy RL formulation. To integrate diffusion sampling with RL objectives, the authors reformulate the problem as an augmented MDP, where each diffusion step is treated as part of the decision-making process. Based on this framework, the paper introduces diffusion-based variants of several existing RL algorithms, including DME-PPO (diffusion PPO), DME-REPPO (diffusion relative entropy PPO), and DME-WPO (diffusion Wasserstein policy optimization). Experiments on continuous-control benchmarks suggest that these variants often outperform their corresponding base algorithms.

**Compliance With Llm Reviewing Policy:**

Affirmed.

**Final Justification:**

As some of my concerns have not been resolved, I will keep my score.

**Key Questions For Authors:**

1.	The augmented diffusion MDP significantly increases complexity. Is there a simpler formulation that avoids introducing diffusion steps as MDP states?
2.	The diffusion policy is sampled without value guidance. Could the authors discuss whether incorporating value-guided diffusion sampling (e.g., Q-guidance) would further improve performance or reduce the number of diffusion steps?
3.	The proposed framework introduces an augmented diffusion MDP where diffusion steps are treated as environment transitions. However, since the diffusion process is internal to the policy and the environment state remains fixed during these steps, it is unclear whether this reformulation is strictly necessary. Could the authors clarify whether similar training objectives could be derived without redefining the MDP, and what practical advantages the diffusion MDP formulation provides?
4.	Diffusion policies are motivated by their ability to model complex and multimodal action distributions. However, the experiments focus on standard continuous control benchmarks where optimal policies are often unimodal. Could the authors provide experiments or analysis demonstrating scenarios where the additional expressiveness of diffusion policies provides a clear advantage?
5.	Could the authors provide a more detailed analysis of the performance–compute tradeoff, such as improvements normalized by training time or number of environment interactions?

**Limitations:**

yes

**Strengths And Weaknesses:**

**Strengths**

The paper proposes Diffusion-based Maximum Entropy RL (DME-RL), a general framework that integrates diffusion policies with RL, thereby unifying several diffusion-based RL approaches. This work construct a diffusion MDP in which diffusion steps are treated as intermediate states, and the reward is only applied at the final step. This formulation enables RL training with diffusion-based policies.

**Weaknesses**

1. This paper introduces an augmented time index, diffusion MDP, and multiple state/action transformations, which makes the algorithm difficult to follow. Much of this complexity arises from rewriting the diffusion sampling procedure as an MDP, which may obscure the underlying intuition. Furthermore, the notation in the paper is sometimes confusing; for example, the definition of the unnormalized target distribution in the left column of line 082–087 is not clearly explained.

2. Most components are adaptations of existing methods, and the primary novelty appears to lie mainly in the unified formulation.

3. Although the paper presents a formal framework, it does not clearly address whether diffusion improves exploration or whether it effectively captures multimodal optimal actions.

4. The training of DME-RL requires value function evaluation at each diffusion step, which introduces significant computational overhead. The experiments indicate that the runtime can be up to 5.5$\times$ slower than baseline algorithms.

5. It remains unclear whether the method scales well or provides substantial practical benefits, as the reported improvements in the experimental studies are sometimes modest.

---

> ### Author Rebuttal · Authors · 2026-03-28
>
> We thank the reviewer for the thoughtful and constructive feedback. Below we address the main concerns:
>
> **A1: W3, Q1, Q3 - Why introduce the diffusion MDP at all? Could this be avoided?**
> Yes, diffusion-RL objectives can be derived without explicitly defining an augmented MDP, as in DIME (see [link](https://github.com/dmerlicml/DMERL_Rebuttal/blob/main/Table/DIMEvsDMERL.pdf) for a one-to-one comparison to DME; equations there are denoted T-Eq. X). However, that route leads to objectives over the **entire reverse diffusion chain**; see T-Eqs. (1), (5), and (10). In contrast, our diffusion-MDP formulation yields a **step-wise objective**: optimization is performed at sampled diffusion steps using a diffusion-aware value function, without backpropagating through the full reverse chain; see T-Eqs. (3), (7), and (11). This is why diffusion steps are part of the MDP state. The diffusion-MDP is therefore not just a reformulation; it enables a novel, more flexible RL objective. We agree this should be stated more clearly and will revise accordingly. See also A2 to reviewer k9mC.
>
> **A2: W2 - Is the contribution mainly a unified formulation?**
> We respectfully disagree. The **maximum-entropy diffusion MDP** itself (Eqs. 11–13) is, to our knowledge, new and provides a principled recipe for deriving **diffusion-policy counterparts of max-ent RL algorithms**. From it, we derive **DME-PPO, DME-REPPO, and DME-WPO**, all of which are new. As stated in L.077ff, the only direct connection to prior work is that **DPPO emerges as the zero-temperature special case of DME-PPO**. Thus, the paper does not simply unify existing methods; it introduces a new framework, and new algorithms can be derived from it.
>
> **A3: W3, Q4 - Exploration and multimodality .**
> We added a toy example ([link](https://github.com/dmerlicml/DMERL_Rebuttal/blob/main/MultimodalActions/multi_agents.gif)) showing that our methods can learn **multimodal action distributions**. A 2D agent outputs actions in $[-1,1]$, corresponding to realtive direction changes between $-90$° and $90$°. Rewards are given by a double-well potential with optima at $-45$° and $45$°. The state is the agent’s orientation. The transition maps the chosen action to the nearest global maximum, while the reward is computed from the original action. This yields only 8 states, allowing us to visualize, for each state, the learned action histogram ([link](https://github.com/dmerlicml/DMERL_Rebuttal/blob/main/MultimodalActions/histo.png)) alongside the reward landscape.
>
> We compare **REPPO, DIME-REPPO, DPPO, DME-REPPO, DME-WPO, and DME-PPO**. Due to **REPPO**’s Gaussian policy parametrization, the method can only cover one mode and thus exhibits mode collapse. The diffusion baseline **DPPO** [1] also fails to capture the multimodal action distribution and instead produces cyclic behavior. In contrast, **DIME-REPPO** and our methods **DME-REPPO, DME-WPO, and DME-PPO** cover both reward maxima for each state, resulting in richer behavior. This supports our claim that the proposed framework improves policy expressiveness when multimodality matters.
>
> The toy example also shows that reward alone is insufficient to reveal whether a method captures the full multimodal structure: all methods achieve similar reward, yet diffusion-based methods produce richer behavior, i.e., a higher-entropy trajectory distribution due to better multimodal coverage. We therefore view the practical value of diffusion policies not only in return, but also in their ability to represent richer action distributions than standard unimodal policies.
>
> **A4: Q2 - Q-guided diffusion sampling**
> Following the reviewer’s suggestion, we conducted experiments to evaluate Q-guided diffusion sampling. Results are available at [link](https://github.com/dmerlicml/DMERL_Rebuttal/blob/main/Guidance/Guidance.pdf). To introduce guidance, we incorporated an interpolation parameter, $\alpha$, and a step size, $dt$, to calibrate the gradient magnitude of the Q-function. When $\alpha = 1$, the Q-function serves as the sole control signal. Results indicate that, even with a well-calibrated $dt$, pure guidance via the Q-function generally yields inferior results compared to the learned score. However, we observe significant improvements for intermediate values of $\alpha$ on HopperHop. There, the results suggest that small $\alpha$ values lead to improvements, provided $dt$ is chosen appropriately.
>
> **A5: Q5 - Performance / compute normalization**
> All results in Figures 1 and 2 are already normalized by the number of **environment interactions**. Compute efficiency is important, and we discuss this runtime/compute trade-off in L.411ff and Appendix Tab: 1. We also provide per-algorithm runtime on each environment at [link](https://github.com/dmerlicml/DMERL_Rebuttal/blob/main/Table/RuntimeTable.pdf).
>
> We will incorporate this discussion into the revised manuscript.
>
> [1] Ren, A. Z., et al. *Diffusion Policy Policy Optimization.* ICLR 2025.

---

> > ### Author Rebuttal · Reviewer_MgQ2 · 2026-04-01
> >
> > Thank you for the additional experiments and detailed explanation.  I still have question that not be addressed.
> >
> > + Could the authors clarify the role of the assumption $\pi(a \mid s)=\exp(\alpha R_{env}(s,a))$? In standard maximum-entropy reinforcement learning, the policy is typically defined in terms of a soft action-value function rather than the immediate reward alone, since the quality of an action depends not only on its instantaneous reward but also on its effect on future states. As written, this assumption appears nonstandard for a general MDP. If it is intended only as an intermediate unnormalized trajectory-level construction, this should be stated more explicitly to avoid confusion.
> >
> > + My second concern is about the claimed efficiency advantage of the proposed formulation. The authors emphasize that, unlike DIME / REPPO-DIME, their method does not require backpropagation through the full reverse diffusion chain, since the objective is written at a single sampled diffusion step in the augmented MDP. However, this does not remove the need to learn a diffusion-aware critic/value function over the augmented state space, and the method still requires value-function iteration across diffusion steps. In fact, the paper itself later acknowledges this as a key limitation, and the reported runtime numbers do not suggest an overall computational advantage over REPPO-DIME. Therefore, the current explanation seems to justify mainly a memory-flexibility benefit, rather than a clear efficiency improvement in total computation. I encourage the authors to clarify this distinction more explicitly.
> >
> > +  I am not yet convinced about the practical significance of the new derivation. In particular, diffusion policies can already be incorporated into existing RL algorithms more directly, so the paper needs to clarify what is gained beyond a more formal rederivation. As I understand it, the main value of the framework is conceptual: it provides a reverse-KL-based maximum-entropy interpretation, an augmented diffusion MDP, and a systematic recipe for deriving diffusion-policy variants of existing algorithms. However, this does not by itself establish a clear practical advantage over directly combining diffusion policies with standard RL objectives. Since the method still requires diffusion-aware value-function or Q-function evaluation across diffusion steps, the practical benefit of the derivation remains somewhat unclear to me. I therefore encourage the authors to better explain why this framework is materially preferable to a more direct plug-in use of diffusion policies in existing RL methods.

---

> > > ### Author Response · Authors · 2026-04-02
> > >
> > > We apologize that we were unable to address all of the reviewer’s previous questions in our first rebuttal due to the character limit. We appreciate the opportunity to clarify these points here.
> > >
> > > **A1: Role of the unnormalized target distribution in L.82-87**
> > >
> > > The unnormalized target distribution is
> > > $\tilde{\pi}(a_{0:T}) = \int_{s_{0:T+1}} \prod_{t=0}^{T} p(s_{t+1}| s_t,a_t) \tilde{\pi}(a_t| s_t) p(s_0) d s_{0:T+1},$ where $\tilde{\pi}(a_t| s_t) \propto \exp(\alpha R_\text{env}(s_t,a_t)).$
> > > This can be rewritten as $\tilde{\pi}(a_{0:T}) =  \int_{s_{0:T+1}} (p(s_0) \prod_{t=0}^{T} p(s_{t+1}| s_t,a_t) ) \exp(\alpha \sum_{t=0}^{T} R_\text{env}(s_t,a_t)) ds_{0:T+1}.$
> > > As explained in L.88, this is a reward-weighted trajectory distribution: trajectories with higher cumulative reward receive exponentially more probability mass. The weighting is by $\prod_{t=0}^{T} \tilde{\pi}(a_t| s_t)=\exp(\alpha \sum_{t=0}^{T} R_\text{env}(s_t,a_t)),$ not only by the instantaneous term $\exp(\alpha R_\text{env}(s_t,a_t))$ as suggested in the review comment. We will clarify this in our revision.
> > >
> > > **A2: Role of the assumption that $\tilde{\pi}(a_t|s_t) \propto \exp(\alpha R_\text{env}(s_t,a_t))$**
> > > This is a common intermediate assumption in variational-inference views of RL. In particular, it is closely related to Eq. 8 and the equation above Eq. 11 in [1], where the trajectory is weighted by $\exp(\sum_{t=1}^{T} r(s_t,a_t))$. In our notation,
> > > $\prod_{t=1}^{T} \tilde{\pi}(a_t|s_t) \propto \exp(\alpha \sum_{t=1}^{T} R_\text{env}(s_t,a_t)).$
> > >
> > > Thus, we fully agree with the reviewer that the optimal policy is not characterized purely by immediate reward.
> > > It is important to note, however, that as stated in L.100ff, the unnormalized target is intractable and is not the final object being optimized. Instead, we apply the data processing inequality to Eq. 1, yielding the tractable reverse-KL between the joint target and variational distributions in L.110-117, directly analogous in spirit to Eq. 11 in [1]. Applying the policy gradient theorem then gives the surrogate loss in Eq. 4 (L.148ff), which aligns with maximum-entropy RL.
> > > We therefore agree that the optimal policy $q^*(a_t|s_t)$ of the surrogate loss is proportional to $\exp(\alpha Q(s_t,a_t))$, i.e., it depends on future states through the Q-function and not only on immediate reward. This should not be confused with our $\tilde{\pi}(a_t|s_t)$, which plays the role of $p(\mathcal{O}_t |s_t,a_t)$ in Eq. 3 of [1]. Unlike [1], we do not introduce optimality variables because we derive the surrogate loss via the data processing inequality, which is more convenient for deriving the diffusion-based maximum-entropy RL MDP.
> > >
> > > We thank the reviewer for pointing out that this distinction should have been made more explicit.
> > >
> > > **A3: Clarification regarding the claimed efficiency advantage**
> > >
> > > We emphasize that we do not claim that our method is more efficient in total wall-clock time. Our claim is instead a memory/flexibility benefit.
> > > As discussed with reviewer k9mC in A2, our formulation writes the objective at a single sampled diffusion step in the augmented MDP, thereby providing additional flexibility in how computation is distributed across diffusion steps. This yields a memory/runtime trade-off rather than a runtime improvement. Concretely, the inference cost of our loss is $O(1)$ in the number of diffusion steps used in one update, and the memory cost is $O(\kappa)$. However, if $\kappa < K$, one requires $K/\kappa$ more update steps, so there is no overall time-efficiency improvement.
> > > As also noted in L.422ff, the longer runtime in our experiments is partially due to our choice of $\kappa = K/2$ to explicitly demonstrate the diffusion-step subsampling feature.
> > >
> > > We will make sure to state this compute/memory trade-off clearly in the revised manuscript.
> > >
> > > **A4: Practical significance**
> > >
> > > We believe that the practical benefit is partially addressed in our response A2 to reviewer k9mC.
> > > The diffusion-based maximum-entropy MDP yields a formulation with additional memory flexibility while matching REPPO-DIME performance.
> > > Our framework also allows us to combine diffusion policies with PPO without requiring importance weights over the entire reverse diffusion process between the old and current policy, which would likely lead to high-variance updates. Instead, within our framework, the importance weights are defined over a single reverse diffusion step.
> > > Moreover, to the best of our knowledge, the only diffusion-RL method with such flexibility is DPPO, corresponding to DME-PPO at zero temperature. However, as shown in A3 of our first rebuttal, DPPO fails to model multimodal action distributions even on simple toy tasks, which makes the temperature extension practically meaningful.
> > >
> > > We thank the reviewer again for the useful comments and will incorporate this feedback into the revised manuscript.
> > >
> > > [1] Levine, Sergey. *Reinforcement Learning and Control as Probabilistic Inference: Tutorial and Review*

---

### Official Review · Reviewer_k9mC · 2026-03-10

**Soundness:** 3
**Presentation:** 2
**Significance:** 2
**Originality:** 3
**Overall Recommendation:** 4
**Confidence:** 3

**Summary:**

This paper introduces a maximum entropy reinforcement learning framework for diffusion-based policies. The proposed method introduces a lower bound to the KL divergence that follows from framing reinforcement learning through the lens of probabilistic inference. The resulting bound is tractable and allows for optimizing the parameters of the diffusion policy. Additionally, the proposed method considers a Q-function that is aware of the latent actions that follow from the diffusion process. On varying simulated control benchmarks, it is shown that the proposed method can perform well.

**Compliance With Llm Reviewing Policy:**

Affirmed.

**Final Justification:**

I have adjusted my score from 3 to 4. I still believe the paper needs major presentation improvements.

**Key Questions For Authors:**

- What exactly are the differences to DIME [1]? When I skim their equations, it looks very similar to what I see in this work.

- Is REPPO-DIME a variant of DIME? In this case, the paper states that DIME requires backpropagating through the whole diffusion chain, but it seems that REPPO-DIME's performance is similar to the proposed method here, while being faster in terms of run time according to the reported values in the appendix. Where is the bottleneck in terms of the runtime? What is the benefit of not backpropagating through the chain here?

- The paper introduces an extended MDP that incorporates the diffusion process's MDP and defines an augmented reward (Eq.13), which boils down to the environment reward, as there are no rewards during the denoising process. In this case, why exactly is introducing the MDP necessary?

- Also related to the previous question, I don't understand the justification for why the Q-function depends on the latent actions at all? How is this mathematically justified?

- Related to the previous question, the paper does not state how the Q-function is learned in general. Given that it depends on latent actions, this needs clarification. Additionally, how is the entropy term handled, as the paper correctly states that the marginal distribution of the policy is intractable?

Overall, this paper looks promising to me, but I have concerns about the presentation, which makes it hard to assess.

[1] Celik O., et al. Diffusion-Based Maximum Entropy Reinforcement Learning. ICML 2025.

**Limitations:**

yes

**Strengths And Weaknesses:**

Strengths:

- The paper tackles a relevant problem in reinforcement learning

- Although I have some concerns regarding the presentation, in general, the paper is well-written, such that the reader can easily follow the ideas and motivation


Weaknesses:

- While I appreciate the long and detailed problem description (Section 2) for clarity, I believe relative to the method's text, it is too long, such that the paper does not provide the necessary information to understand the proposed method's details (see questions). For example, the introduction of Section 2 and Section 2.1 are well-known in the maximum entropy RL literature and can be formulated in a more compact format. Similarly, although the connection to the log variance loss is interesting, I don't really understand why it is relevant to discuss the connection here.


- The paper lacks a related work section that distinguishes itself clearly from prior works in the main text. For example, the paper's title and the algorithm's name reads very similarly to a prior work [1]. It should be clearly clarified what exactly the differences are, so the reader can easily assess the tackled problem and the contributions. While information is provided in the Appendix, it is important to provide the most important differences in a compact way in the main text.

- The paper lacks comparison to other on-policy flow/diffusion-based RL methods [2,3]

- Minor, but important for readers to assess the performance: The paper reports 7 independent seeds for the tasks, but in RL, it is generally important to run on more seeds, and it has been shown beneficial to report the interquartile mean with 95% bootstrapped confidence intervals as reported in [4]. This has become common in the RL community. I highly recommend following the procedure.

- Please also see the question for the authors part

[1] Celik O., et al. Diffusion-Based Maximum Entropy Reinforcement Learning. ICML 2025.

[2] Ren A. Z., et al. Diffusion Policy Policy Optimization. ICLR 2025. ICLR 2025.

[3] McAllister D. et al. Flow Matching Policy Gradients. 2025.

[4] Agarwal, R. et al. Deep reinforcement learning at the edge of the statistical precipice. NeurIPS 2021.

---

> ### Author Rebuttal · Authors · 2026-03-28
>
> We thank the reviewer for the careful reading and constructive feedback. Below we address the main concerns and clarify the relation to prior work. At [this link](https://github.com/dmerlicml/DMERL_Rebuttal/blob/main/Table/DIMEvsDMERL.pdf), we provide a one-to-one comparison between the equations of REPPO-DIME and DME-REPPO; we refer to these as (T-Eq. X).
>
> **A1: W1 - Length of Sec. 2.**
> We agree that Secs. 2 and 2.1 can be condensed. Our goal was to make the paper self-contained and establish notation, also this section is not purely background: unlike the standard derivation used in max-ent RL, we rely on the data processing inequality (DPI), which is convenient for deriving the MaxEnt diffusion-MDP formulation.
>
> **A2: W2, Q1, Q2 - Difference to REPPO-DIME.**
> We agree that the distinction to DIME should be stated clearly in the main text rather than mainly in the appendix.
> The key difference is that DIME / REPPO-DIME optimizes objectives over the **full reverse diffusion trajectory**. This is visible in the policy loss and KL regularizer, which are defined over $a_t^{0:K}$ in (T-Eqs. 1, 5). Both losses, therefore, require simulating the full reverse diffusion chain, leading to $\mathcal{O}(K)$ time and memory costs. In contrast, our method defines both losses at a **single reverse diffusion step** conditioned on the augmented state $\tilde s_{\tilde t}=(s_t,a_t^k,k)$; see (T-Eqs. 3, 7). This enables diffusion-step subsampling and yields $\mathcal{O}(1)$ cost per sampled step and $\mathcal{O}(\kappa)$ memory for a minibatch of $\kappa$ sampled steps. However, if we train on all rollout data, we perform roughly $K/\kappa$ more updates, allowing our methods to trade off runtime compute and memory costs. The step-wise formulation becomes increasingly beneficial for larger policies, longer diffusion horizons, or settings such as fine-tuning large diffusion policies, where full-chain backpropagation is costly.
>
> **A3: Q3 - Comparison to other on-policy diffusion/flow RL methods.**
> We agree that a broader comparison would strengthen the paper. At the same time, we already compare against the diffusion-policy baseline REPPO-DIME in Figs. 1 and 2. Moreover, as explained in L.076ff, DME-PPO reduces to DPPO at temperature 0, so DPPO is a special case of our formulation, while our method generalizes it to the max-ent RL setting. We also refer to our answer A1 to Reviewer E8RG, where we added a toy experiment showing that our method can cover multimodal action distributions, whereas DPPO fails.
>
> **A4: Q4 - Why does the Q-function depend on latent actions?**
> We justify this in Sec. 3. Starting from Eq. 8, we apply the DPI to obtain a tractable upper bound for diffusion policies. From Eq. 10 to Eq. 11, we then apply the policy gradient theorem (derived in App. G), which yields the surrogate loss in Eq. 11 together with the value- and Q-function definitions in Eq. 12. Intuitively, under a diffusion policy, the joint reverse process decomposes into a product of reverse diffusion steps. Each reverse step can therefore be interpreted as a noisy intermediate policy, so the resulting Q-function must depend on latent actions.
>
> **A5: Q3 - Why is the diffusion MDP necessary?**
> The diffusion MDP is a direct consequence of applying the policy gradient theorem to Eq. 10. While the reward reduces to the environment reward at the last diffusion step, the value function additionally contains the log-ratios between forward and reverse diffusion transitions (see Eq. 12 and L.280ff). Since $Q = R + \mathbb{E}[V]$, one could equivalently absorb this log-ratio term into the reward, as in App. G, Eq. 39. We chose the present formulation because in max-ent RL the entropy term is usually incorporated into the value function, and we mirror that convention here. The augmented diffusion-MDP also enables step-wise optimization and thus diffusion-step subsampling.
>
> **A6: Q5 - How is the Q-function learned?**
> We agree this should be explained more clearly. In the linked table above, we provide the corresponding equations. In our formulation, the critic is trained on augmented tuples $(\tilde s_{\tilde t}, a_t^{k-1})$ via the loss in (T-Eq. 14), with TD-$\lambda$-style targets in (T-Eqs. 17, 18). This is the DME-REPPO analogue of the REPPO-DIME critic loss in (T-Eq. 13) with targets in (T-Eqs. 15, 16). We will make this clearer in the revision and add pseudocode.
>
> **A7: Q5 - How is the intractable entropy term handled?**
> As explained around L.231ff (right), applying the DPI to the intractable KL yields an upper bound in which all terms are tractable; in particular, the marginal distribution is no longer required.
>
> **A8: W4 - Statistical reporting.**
> At [this link](https://github.com/dmerlicml/DMERL_Rebuttal/blob/main/IQM/all_envs_methods_iqm_eval_return.png), the reviewer can find updated figures with IQM results.
>
> We thank the reviewer again for the helpful comments. We will incorporate these clarifications in the revised manuscript.

---

> > ### Author Rebuttal · Reviewer_k9mC · 2026-04-01
> >
> > I would like to thank the authors for their detailed responses and their efforts.
> >
> > I will adjust my score. However, I still think that the paper needs an improved presentation as mentioned in my original review.

---

> > > ### Author Response · Authors · 2026-04-02
> > >
> > > We thank the reviewer again for the helpful feedback and for raising the score.
> > >
> > > We agree that the presentation can be improved, and we will use the revision to make the method easier to follow. In particular, we will shorten the background discussion, make the distinction to previous works such as DIME / REPPO-DIME more explicit in the main text, and improve the explanation of the diffusion-MDP and diffusion-aware critic losses.
> > >
> > > We will also add pseudocode for **DME-WPO**, **DME-PPO**, and **DME-REPPO**, and include a concise one-to-one comparison between **DME-REPPO** and **REPPO-DIME**, following the structure demonstrated in the linked table. Our goal is to make the algorithmic differences and implementation details easier to assess directly from the main paper.

---

### Decision · Program_Chairs · 2026-04-30

**Decision:**

Reject

**Comment:**

The paper proposes DME-RL (Diffusion-Based Maximum Entropy Reinforcement Learning), a framework that frames ME-RL as a diffusion-based sampling problem. By introducing an augmented "Diffusion MDP," the authors derive diffusion-based variants of existing RL algorithms, including DME-PPO, DME-REPPO, and DME-WPO.

## Strengths:

- Unified framework: Reviewers appreciated the conceptual elegance of formulating a Diffusion MDP that can be integrated into various established RL algorithms.

- Memory efficiency: Compared to baselines that require backpropagation through the entire reverse diffusion chain (e.g., DIME), the proposed step-wise optimization offers valuable memory flexibility.

- Multimodal expressiveness: The toy experiment added during the rebuttal successfully demonstrated the framework's ability to capture multimodal action distributions, overcoming the mode-collapse seen in standard Gaussian policies.

## Key Weaknesses & Unresolved Issues:

Despite the authors' commendable efforts during the rebuttal to clarify differences from prior work and provide additional experiments, several critical issues remain unresolved:

**Lack of theoretical rigor** Reviewer o2Q3 raised fundamental concerns regarding the mathematical soundness of the derivations. Specifically, applying the classical policy gradient theorem to an objective containing parameter-dependent log-ratios acts as a tailored surrogate rather than a direct mathematical equivalence. Furthermore, the derivations handling learnable forward/reverse kernels and the Wasserstein gradient projection lack the necessary rigor and explicit assumptions in the main text. The rebuttal did not fully resolve these foundational concerns.

**High computational overhead vs. marginal gains** As highlighted by Reviewer MgQ2, the method requires value-function evaluations at every diffusion step. This results in a massive computational overhead (running 4x to 5.5x slower than baselines) while yielding only modest performance improvements on standard continuous control benchmarks. The practical utility and scalability of the method are therefore highly questionable.

**Presentation and clarity** Multiple reviewers (k9mC, MgQ2) noted that the paper spends an excessive amount of space on standard background material at the expense of methodological clarity. The core differences between this work and closely related concurrent methods were not made sufficiently clear in the main text prior to the rebuttal.

While the concept of a Diffusion MDP is innovative and shows potential for handling multimodal action spaces, the current manuscript suffers from theoretical inaccuracies and a poor compute-to-performance trade-off. A substantial revision is needed to rigorously formalize the mathematical claims, streamline the presentation, and better justify the heavy computational cost.